# Translatome analysis reveals cellular network in DLK-dependent hippocampal glutamatergic neuron degeneration

**Erin M Ritchie[1,2], Dilan Acar[1], Siming Zhong[1], Qianyi Pu[1], Yunbo Li[1], Binhai Zheng[3], Yishi Jin[1,3,4]***

[1]Department of Neurobiology, School of Biological Sciences, University of California San Diego, La Jolla, United States; [2]Biomedical Sciences Graduate Program, School of Medicine, University of California San Diego, La Jolla, United States; [3]Department of Neurosciences, School of Medicine, University of California San Diego, La Jolla, United States; [4]Kavli Institute for Brain and Mind, University of California San Diego, La Jolla, United States

## eLife Assessment

This manuscript describes the impact of modulating signaling by a key regulatory enzyme, Dual Leucine Zipper Kinase (DLK), on hippocampal neurons. The results are interesting and will be **important** for scientists interested in synapse formation, axon specification, and cell death. The authors have carefully addressed the comments made by the reviewers and the findings are **convincing** in large part due to the use of extensive mouse genetics, detailed gene expression of enriched genes, and recognition of neuron vulnerability.

**\*For correspondence:**
yijin@ucsd.edu

**Abstract** The conserved MAP3K12/Dual Leucine Zipper Kinase (DLK) plays versatile roles in neuronal development, axon injury and stress responses, and neurodegeneration, depending on cell-type and cellular contexts. Emerging evidence implicates abnormal DLK signaling in several neurodegenerative diseases. However, our understanding of the DLK-dependent gene network in the central nervous system remains limited. Here, we investigated the roles of DLK in hippocampal glutamatergic neurons using conditional knockout and induced overexpression mice. We found that dorsal CA1 and dentate gyrus neurons are vulnerable to elevated expression of DLK, while CA3 neurons appear less vulnerable. We identified the DLK-dependent translatome that includes conserved molecular signatures and displays cell-type specificity. Increasing DLK signaling is associated with disruptions to microtubules, potentially involving STMN4. Additionally, primary cultured hippocampal neurons expressing different levels of DLK show altered neurite outgrowth, axon specification, and synapse formation. The identification of translational targets of DLK in hippocampal glutamatergic neurons has relevance to our understanding of selective neuron vulnerability under stress and pathological conditions.

## Introduction

The mammalian Mitogen Activated Protein Kinase Kinase Kinase (MAP3K12) Dual Leucine Zipper Kinase (DLK) is broadly expressed in the nervous system from early development to mature adults. DLK exerts its effects primarily through signal transduction cascades involving downstream MAP2Ks (MKK4, MKK7) and MAPKs (JNK, p38, ERK), which then phosphorylate many proteins, such as transcription factors including c-Jun, to regulate cellular responses (*Asghari Adib et al., 2018*; *Hirai*

*et al., 2006*; *Huang et al., 2017*; *Itoh et al., 2009*; *Jin and Zheng, 2019*; *Tedeschi and Bradke, 2013*). In cultured neurons, DLK is localized to axons (*Hirai et al., 2005*; *Lewcock et al., 2007*), dendrites (*Pozniak et al., 2013*), and the Golgi apparatus (*Hirai et al., 2002*). DLK is also associated with transporting vesicles, which are considered platforms for DLK to serve as a sensor of neuronal stress or injury (*Holland et al., 2016*; *Tortosa et al., 2022*). Despite broad expression, functional investigations of DLK have been limited to a few cell types under specific conditions.

Constitutive DLK knockout (KO) mice, generated by removing the N-terminus of DLK, including the ATP binding motif of the kinase domain (*Hirai et al., 2006*), or by deleting the entire kinase domain (*Ghosh et al., 2011*), die perinatally. The development of the embryonic nervous system is largely normal, with mild defects in radial migration and axon track formation in the developing cortex (*Hirai et al., 2006*) and neuronal apoptosis during development of spinal motor neurons and dorsal root ganglion (DRG) neurons (*Ghosh et al., 2011*; *Itoh et al., 2011*). Selective removal of DLK in layer 2/3 cortical neurons starting at E16.5 results in increased dendritic spine volume (*Pozniak et al., 2013*). Induced deletion of DLK in adult mice causes no obvious brain structural defects; synapse size and density in hippocampus and cortex appear unaltered, although basal synaptic strength is mildly increased (*Pozniak et al., 2013*). In contrast, under injury or stress conditions, DLK exhibits critical context-specific roles. In DRG neurons, DLK is required for nerve growth factor withdrawal induced death, promotes neurite regrowth, and is also involved in retrograde injury signaling (*Ghosh et al., 2011*; *Holland et al., 2016*; *Itoh et al., 2009*; *Shin et al., 2012*). In a spinal cord injury model, DLK is required for *Pten* deletion-induced axon regeneration and sprouting as well as spontaneous sprouting of uninjured corticospinal tract neurons (*Saikia et al., 2022*). In optic nerve crush assay, DLK is necessary for *Pten* deletion-induced axon regeneration of retinal ganglion cells (RGC), but also contributes to injury-induced RGC death (*Watkins et al., 2013*). In a mouse model of stroke, increased DLK expression is associated with motor recovery following knockdown of the CCR5 chemokine receptor (*Joy et al., 2019*). These studies reveal critical roles of DLK in development, maintenance, and repair of neuronal circuits.

DLK is known to be expressed in hippocampal neurons (*Blouin et al., 1996*; *Hirai et al., 2005*; *Mata et al., 1996*). Loss of DLK, either constitutively or in adult animals, causes no discernable effect on hippocampal morphology (*Hirai et al., 2006*; *Pozniak et al., 2013*). Microarray-based gene expression analysis did not detect significant changes associated with loss of DLK in the hippocampus (*Pozniak et al., 2013*). However, following exposure to kainic acid, loss of DLK, or preventing phosphorylation of the downstream transcription factor c-Jun, significantly reduces neuron death in hippocampus (*Behrens et al., 1999*; *Pozniak et al., 2013*). Additionally, elevated levels of p-c-Jun are observed in hippocampus of patients with Alzheimer's disease (*Le Pichon et al., 2017*). Induced human neurons treated with ApoE4, a prevalent ApoE variant associated with Alzheimer's disease, also show upregulation of DLK, which leads to enhanced transcription of APP and thus Aβ levels (*Huang et al., 2017*). These data suggest that transcriptional changes downstream of DLK may be an important aspect of its signaling in hippocampal neuron degeneration under pathological conditions.

Here, we investigate the DLK-dependent molecular and cellular network in hippocampal glutamatergic neurons, which show selective vulnerability in Alzheimer's disease, ischemic stroke, and excitotoxic injury. Using DLK conditional knockout and overexpression mice, we reveal hippocampal regional differences in neuronal death upon elevated DLK signaling. We describe translational changes in hippocampal glutamatergic neurons using RiboTag-seq analysis. We show that the key transcription factor c-Jun and a member of the stathmin family, STMN4, display DLK-dependent translation. Our analyses on hippocampal tissues and cultured neurons support the conclusion that the DLK-dependent signaling network has important roles in the regulation of microtubule homeostasis, neuritogenesis, and synapse formation.

## Results

### DLK conditional knockout in differentiating and mature glutamatergic neurons does not alter gross morphology of hippocampus

As a first step to define the roles of DLK in hippocampal glutamatergic neurons, we verified DLK expression (encoded by *Map3k12*) in hippocampal tissue by RNAscope analysis. We observed strong signals in the glutamatergic pyramidal cells and granule cells in P15 mice (*Figure 1—figure*

supplement 1A), consistent with prior in situ data (*Blouin et al., 1996*; *Lein et al., 2007*; *Mata et al., 1996*). To selectively delete DLK in glutamatergic neurons, we generated *Slc17a7^Cre/+*;*Map3k12^fl/fl* mice (DLK(cKO)). *Map3k12^fl/fl* have LoxP sites flanking the exon encoding the initiation ATG and the first 149 amino acids (*Figure 1—figure supplement 1C*; *Chen et al., 2016*; *Li et al., 2021*; *Saikia et al., 2022*). In hippocampus *Slc17a7^Cre* (encoding VGLUT1) express Cre recombinase strongly in CA3 and in a subset of pyramidal neurons close to stratum oriens in CA1 at P4, with broad expression in both CA1 and CA3 by P14 (*Harris et al., 2014*). In dentate gyrus, expression of *Slc17a7^Cre* begins in neurons nearer the molecular layer around P4, with expression spreading towards the polymorph layer gradually during the first two postnatal months. By western blot analysis of hippocampal protein extracts, we found that full-length DLK protein was significantly reduced in DLK(cKO) (*Figure 1A and B*). The DLK antibody also detected a protein product of lower molecular weight at much reduced levels (*Figure 1—figure supplement 1B*). As *Map3k12* mRNA lacking the floxed exon was expressed (*Figure 1—figure supplement 1C*), this lower-molecular weight protein could be produced by using a downstream alternative start codon (*Figure 1—figure supplement 1C1*), but would lack the N-terminal palmitoylation motif and ATP-binding site that are essential for DLK activity (*Holland et al., 2016*; *Huntwork-Rodriguez et al., 2013*). These data provide validation for knockout of functional DLK protein in hippocampal glutamatergic neurons in DLK(cKO) mice.

The DLK(cKO) mice were indistinguishable from control littermate mice in behavior and appearance from birth to about one year of age. We examined tissue sections of hippocampus in P15 and P60 mice. Hippocampal sections stained with NeuN, a marker of neuronal nuclei (*Mullen et al., 1992*), showed no significant difference in overall position of neuronal soma or thickness of the CA1 pyramidal cell layer at either timepoint (*Figure 1C and D*). Neuronal morphology visualized by immunostaining with Tuj1, labeling neuron-specific β-III tubulin, also showed no detectable differences in the pattern and intensity of microtubules in DLK(cKO) mice, compared to control (*Figure 1E–H*). Gross morphology of hippocampus and surrounding tissues in 1-year-old DLK(cKO) mice was indistinguishable from controls (*Figure 1—figure supplement 1E, G-I*, *Figure 1—figure supplement 2A*). These results show that DLK does not have essential roles in post-mitotic hippocampal glutamatergic neuron maintenance.

## Increasing expression levels of DLK leads to hippocampal neuron death, with dorsal CA1 neurons showing selective vulnerability

Several studies have reported that DLK protein levels increase under a variety of conditions, including optic nerve crush (*Watkins et al., 2013*), NGF withdrawal (~twofold; *Huntwork-Rodriguez et al., 2013*; *Larhammar et al., 2017*), and sciatic nerve injury (*Larhammar et al., 2017*). Induced human neurons show increased DLK abundance about ~fourfold in response to ApoE4 treatment (*Huang et al., 2019*). Increased expression of DLK can lead to its activation through dimerization and autophosphorylation (*Nihalani et al., 2000*). We thus asked how increased DLK signaling affects hippocampal glutamatergic neurons. We previously described a transgenic mouse, H11-DLK(iOE), which allows Cre-dependent DLK overexpression (*Li et al., 2021*). The DLK transgene is coexpressed with tdTomato through a T2A peptide (*Figure 1—figure supplement 1D*). By RNAscope analysis, compared to control, we observed increased *Map3k12* mRNAs in glutamatergic neurons in CA1, CA3, and DG at P15 in *Slc17a7^Cre/+*;H11-DLK(iOE)/+ mice (referred to as DLK(iOE)) (*Figure 2—figure supplement 1A and B*). By immunostaining hippocampal sections with anti-DLK antibodies, we observed increased protein levels particularly in regions with pyramidal neuron dendrites in DLK(iOE), compared to control mice (*Figure 2—figure supplement 1C–E*). Additional analysis at the mRNA level (*Supplementary file 2* WT vs DLK(iOE) DEGs) and at the protein level (*Figure 4—figure supplement 1E*) suggest that the increase in DLK abundance was around three times the control level. The localization patterns of DLK protein appeared to vary depending on region of hippocampus and age of animals in both control and DLK(iOE) mice (*Figure 2—figure supplement 1C*).

DLK(iOE) mice were born normally, and developed noticeable progressive motor deficits around four months of age, which became worse by one year of age. We stained brain sections for NeuN at P10, P15, P60, and 1 year of age and observed a progressive reduction in brain size of these mice, compared to controls (*Figure 1—figure supplement 2B and C*). At P10, the dorsal hippocampus in DLK(iOE) was indistinguishable from control (*Figure 2A, B*, *Figure 1—figure supplement 2B*). By P15, the DLK(iOE) mice showed significant thinning of the CA1 pyramidal layer. We detected increased

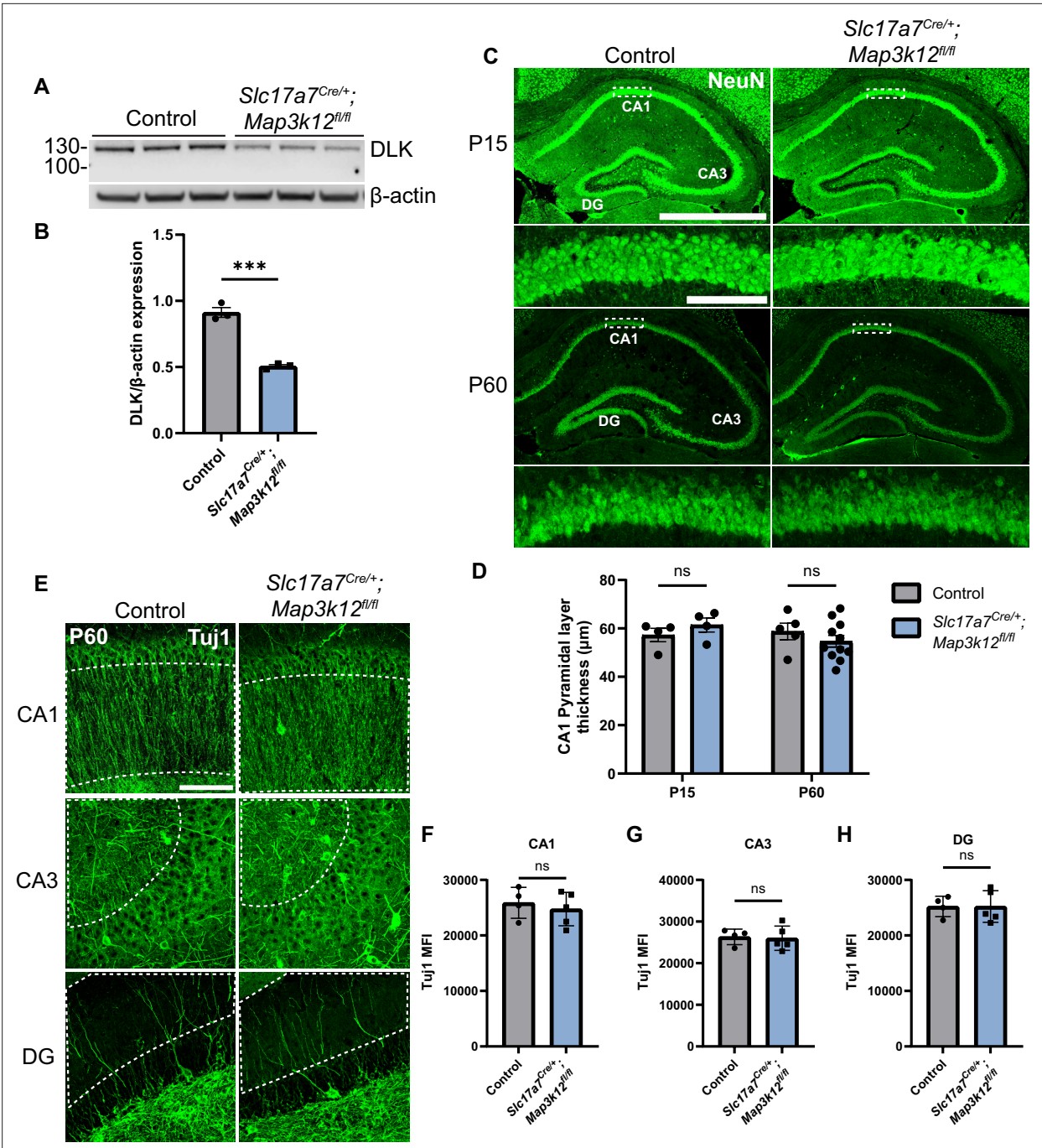

**Figure 1.** Deletion of DLK in postmitotic glutamatergic neurons does not alter gross morphology of hippocampus. (**A**) Western blot of DLK and β-actin in protein extracts of hippocampal tissue of *Slc17a7^{Cre/+}*;*Map3k12^{fl/fl}* and *Map3k12^{fl/fl}* littermate controls (age P60, each lane representing individual mice, N=3 mice/genotype). (**B**) Quantification of DLK protein level normalized to β-actin. Statistics: Unpaired t-test, *** p<0.001. Error bars represent SEM. (**C**) Confocal z-stack (max projection) images of NeuN immunostaining of coronal sections of the dorsal hippocampus in P15 and P60 mice of genotype indicated, respectively. Dashed boxes in CA1 pyramidal layers are enlarged below. Scale bar, 1000 µm in hippocampi; 100 µm in CA1 layer. (**D**) Quantification of CA1 pyramidal layer thickness. Each dot represents averaged thickness from 3 sections per mouse; N≥4 mice/genotype per timepoint. Statistics: Two-way ANOVA with Holm-Sidak multiple comparison test; ns, not significant. Error bars represent SEM. (**E**) Confocal z-stack (max projection) images of Tuj1 immunostaining of hippocampus CA1, CA3, and DG regions in control and *Slc17a7^{Cre/+}*;*Map3k12^{fl/fl}* mice (age P60). Dashed outlines mark ROI (region of interest) for fluorescence intensity quantification. Scale bar, 100 µm. (**F, G, H**) Tuj1 mean fluorescence intensity (MFI) after thresholding signals in dendritic regions in each hippocampal area. Each dot represents averaged intensity from 3 sections per mouse; N=4 control, 5 *Slc17a7^{Cre/+}*;*Map3k12^{fl/fl}*. Statistics: Unpaired t-test. ns, not significant. Error bars represent SEM.

The online version of this article includes the following source data and figure supplement(s) for figure 1:

*Figure 1 continued on next page*

*Figure 1 continued*

**Source data 1.** Original western blots for images shown in *Figure 1A*.

**Source data 2.** PDF showing original western blots for images shown in *Figure 1A*, along with relevant bands and genotypes.

**Figure supplement 1.** Additional evidence for expression levels of DLK and effects on hippocampal morphology at 1 year of age.

**Figure supplement 1—source data 1.** Original western blots for images shown in *Figure 1—figure supplement 1B*.

**Figure supplement 1—source data 2.** PDF showing original western blots for images shown in *Figure 1—figure supplement 1B*, along with relevant bands and genotypes.

**Figure supplement 2.** Hemibrain images across timepoints.

TUNEL staining signals in CA1 pyramidal layer, compared to control (*Figure 2—figure supplement 3F and G*). By P60, most CA1 pyramidal neurons were lost, while DG began to show thinning, which continued to worsen at 1 year of age (*Figure 2A, B*, *Figure 1—figure supplement 1F, I*, *Figure 1— figure supplement 2B and C*). In contrast, neurons in CA3 appeared less affected, even at 1 year of age (*Figure 2A*, *Figure 1—figure supplement 1F, H*, *Figure 1—figure supplement 2C*). Additionally, in P60, dorsal CA1 showed significantly fewer surviving neurons, while ventral CA1 pyramidal layer thickness appeared more similar to control than dorsal regions (*Figure 2—figure supplement 2A and B*). Neuronal death generally induces reactive astrogliosis. We stained for GFAP, a marker of astrocyte reactivity. We found increased GFAP staining in DLK(iOE), specifically in CA1 at P15, and at P60 in CA1 and DG, but not as strongly in CA3, compared to control mice (*Figure 2—figure supplement 3A–D*). We also stained for IBA1, a marker of microglia, and found that DLK(iOE) mice showed increased IBA1 staining around the CA1 region, compared to control mice (*Figure 2—figure supplement 3E*). Microglia appeared ramified in control mice and more reactive-looking in DLK(iOE) mice (*Figure 2— figure supplement 3E*). Together, these data reveal that dorsal CA1 neurons show vulnerability to elevated DLK expression, while CA3 neurons appear less vulnerable to DLK overexpression.

## DLK dependent translated genes are enriched in synapse formation and function

To gain understanding of molecular changes associated with DLK expression levels in glutamatergic neurons, we next conducted translating ribosome profiling and RNA sequencing (RiboTag profiling) using *Rpl22^HA^* mice, which enables Cre-dependent expression of an HA tagged RPL22, a component of the ribosome, from its endogenous locus (*Sanz et al., 2009*). We generated *Slc17a7^Cre/+^*;H11- DLK(iOE)/+;*Rpl22^HA/+^*, *Slc17a7^Cre/+^*;*Map3k12^fl/fl^*;*Rpl22^HA/+^*, and their respective *Slc17a7^Cre/+^*;*Rpl22^HA/+^* sibling controls. We made protein extracts from dissected hippocampi of P15 mice, a time point when some CA1 neuron degeneration induced by DLK overexpression was visible. We obtained affinity purified HA-immunoprecipitates with the associated actively translated RNAs (*Figure 3—figure supplement 1A*) (n=3 DLK(iOE)/3 WT, n=4 DLK(cKO)/4 WT) and verified purity of the isolated RNA samples by qRT-PCR (*Figure 3—figure supplement 1B*). We mapped >24 million deep sequencing reads per sample to approximately 14,000 genes. We found 260 genes that were differentially expressed and translated in DLK(iOE) neurons, including 114 up- and 146 down-regulated genes, compared to control (*Figure 3A*, using the cutoff of $p_{adj}$ <0.05, *Supplementary file 2* WT vs DLK(iOE) DEGs). 36 genes showed significant changes in DLK(cKO) neurons, including 12 up- and 24 down-regulated genes, compared to control (*Figure 3B*, $p_{adj}$ <0.05, *Supplementary file 1* WT vs DLK(cKO) DEGs). Among genes with statistically significant changes, 17 were detected in both DLK(cKO) and DLK(iOE) (*Figure 3—figure supplement 1C*), of which 13 were upregulated in DLK(iOE) and downregulated in DLK(cKO), and 3 were downregulated in DLK(iOE) and upregulated in DLK(cKO) (*Figure 3—figure supplement 1D and E*). The most significant differentially expressed genes included *Jun,* encoding the DLK downstream transcription factor c-Jun, *Stmn4,* encoding a member of the Stathmin tubulin- binding protein family, and *Sh2d3c*, encoding a SH2-domain cytoplasmic signaling protein (*Dodelet et al., 1999*; *Vervoort et al., 2007*). One gene, *Slc25a17*, a peroxisomal transporter for cofactors FAD, CoA, and others (*Agrimi et al., 2012*) and broadly implicated in oxidative stress, was upregulated in both DLK(cKO) and DLK(iOE), compared to control, though the relevance of this change may require further investigation. To systematically compare whether DLK regulates the translatome in a coordinated manner, we performed rank-rank hypergeometric overlap (RRHO) analysis (*Plaisier et al., 2010*) on the entire translated mRNAs detected in DLK(iOE) and DLK(cKO). We found that

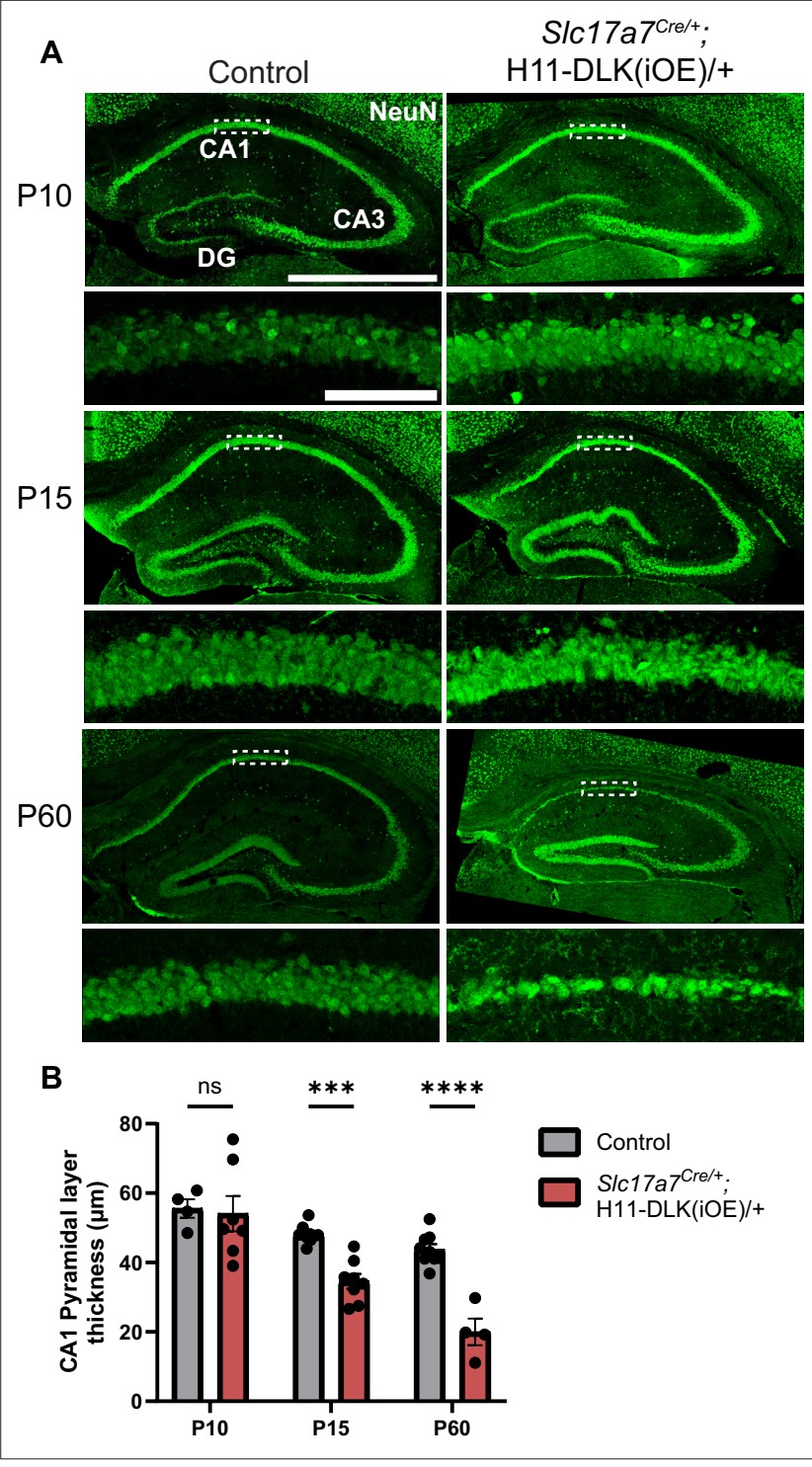

**Figure 2.** Induced DLK overexpression in hippocampal glutamatergic neurons causes degeneration of CA1 neurons. (**A**) Confocal z-stack (max projection) images of NeuN immunostaining of coronal sections from dorsal hippocampus in P10, P15, and P60 mice of genotype indicated. Dashed boxes mark CA1 pyramidal layers enlarged below. P60 images shown under different settings compared to P10 and P15 due to older staining. Scale bar, 1,000 µm in hippocampi; 100 µm in CA1 layer. (**B**) Quantification of CA1 pyramidal layer thickness. Data points represent averaged measurement from 3 sections per mouse, N≥4 mice/genotype at each timepoint. Statistics: Two-way ANOVA with Holm-Sidak multiple comparison test. ns, not significant; *** p<0.001; **** p<0.0001. Error bars represent SEM.

*Figure 2 continued on next page*

*Figure 2 continued*

The online version of this article includes the following figure supplement(s) for figure 2:

**Figure supplement 1.** Evidence for induced DLK expression visualized at RNA and protein levels.

**Figure supplement 2.** Regional vulnerability observed with increased DLK expression.

**Figure supplement 3.** Additional evidence for DLK(iOE) induced hippocampal neuron death.

RRHO detected significant overlap in genes that were upregulated in DLK(iOE) and downregulated in DLK(cKO) as well as the reverse (*Figure 3C*), supporting a conclusion that expression of many of the same genes are dependent on DLK.

To gain understanding of DLK-dependent signaling network, we performed gene ontology (GO) analysis on the 260 genes differentially translated in DLK(iOE) neurons, as this dataset gave greater ability to detect significant GO terms than using the 36 genes differentially expressed in DLK(cKO). The genes upregulated in DLK(iOE) (114) had enrichment in GO terms related to apoptosis, cell migration, cell adhesion, and the extracellular matrix organization (*Figure 3D*), whereas the genes downregulated (146) had GO terms related to synaptic communication and ion transport (*Figure 3E*). Similar GO terms were also identified using the list of genes coordinately regulated by DLK, derived from our RRHO analysis. Among the genes upregulated in DLK(iOE), some were known to be involved in neurite outgrowth (*Plat*, *Tspan7*, *Hap1*), endocytosis or endosomal trafficking (*Snx16*, *Ston2*, *Hap1*), whereas the genes down-regulated in DLK(iOE) included ion channel subunits (*Cacng8*, *Cacng3*, *Grin2b*, *Scn1a*) and those in exocytosis and calcium related proteins (*Doc2b*, *Hpca*, *Cadps2*, *Rab3c*, *Rph3a*). A significant cluster of differentially expressed genes in DLK(iOE) included those that regulate AMPA receptors (*Nptx1*, *Nptxr*, *Cnih3*, *Gpc4*, *Arc*, *Tspan7*) and cell adhesion molecules (*Nectin1*, *Flrt3*, *Pcdh8*, *Plxnd1*). A further survey using SynGO, a curated resource for genes related to synapse formation and function (*Koopmans et al., 2019*), revealed 42 of 260 differentially expressed genes in DLK(iOE) showed significant enrichment in synaptic organization and postsynaptic receptor signaling processes (*Figure 3F*). Conversely, 10 of the 36 differentially expressed genes in DLK(cKO) were annotated to function in similar synaptic processes as in DLK(iOE) (*Figure 3—figure supplement 1J*). The bioinformatic analysis suggests that increased DLK expression can promote translation of genes related to neurite outgrowth and branching and reduce those related to the maturation and function of synapses.

The hippocampus is comprised of multiple glutamatergic neuron types with distinct spatial patterns of gene expression (*Lein et al., 2004*). As we observed regional vulnerability to DLK overexpression, we next asked if the differentially expressed genes associated with DLK(iOE) might show correlation to the neuronal vulnerability. We first surveyed the endogenous expression pattern of the 260 significantly changed genes in DLK(iOE) in hippocampus using in situ data from 8-week-old mice from the Allen Brain Atlas (*Lein et al., 2007*). We found that about a third of the genes downregulated in DLK(iOE) showed enriched expression in CA1 (*Figure 3—figure supplement 1G*), and some of these genes, including *Tenm3*, *Lamp5*, and *Mpped1*, were up-regulated in DLK(cKO) (*Figure 3—figure supplement 1H and I*). In comparison, about 50% of the genes upregulated in DLK(iOE) showed comparable expression among hippocampal cell types (*Figure 3—figure supplement 1F*). Additionally, among the 42 synaptic genes that were differentially expressed in DLK(iOE), a notable portion of the downregulated genes showed enriched expression in CA1 (*Figure 3H*), while the upregulated genes were expressed in all regions (*Figure 3G*).

Additionally, we compared our Slc17a7-RiboTag datasets with CamK2-RiboTag and Grik4-RiboTag datasets from 6-week-old wild type mice reported by *Traunmüller et al., 2023*; GSE209870. We defined a list of genes enriched in CamK2-expressing CA1 neurons relative to Grik4-expressing CA3 neurons (CA1 genes), and those enriched in Grik4-expressing CA3 neurons (CA3 genes) (*Supplementary file 3*). When compared with the entire list of Slc17a7-RiboTag profiling in our control and DLK(cKO), we found CA1 genes tended to be expressed more in DLK(cKO), compared to control (*Figure 3—figure supplement 1K*), while CA3 genes showed a slight enrichment in control though the trend was less significant and less clustered towards one genotype (*Figure 3—figure supplement 1L*). Moreover, many CA1 genes related to cell-type specification, such as *Foxp1*, *Satb2*, *Wfs1*, *Gpr161*, *Adcy8*, *Ndst3*, *Chrna5*, *Ldb2*, *Ptpru*, and *Ntm*, did not show significant downregulation when DLK was overexpressed. These observations imply that DLK likely specifically down-regulates CA1

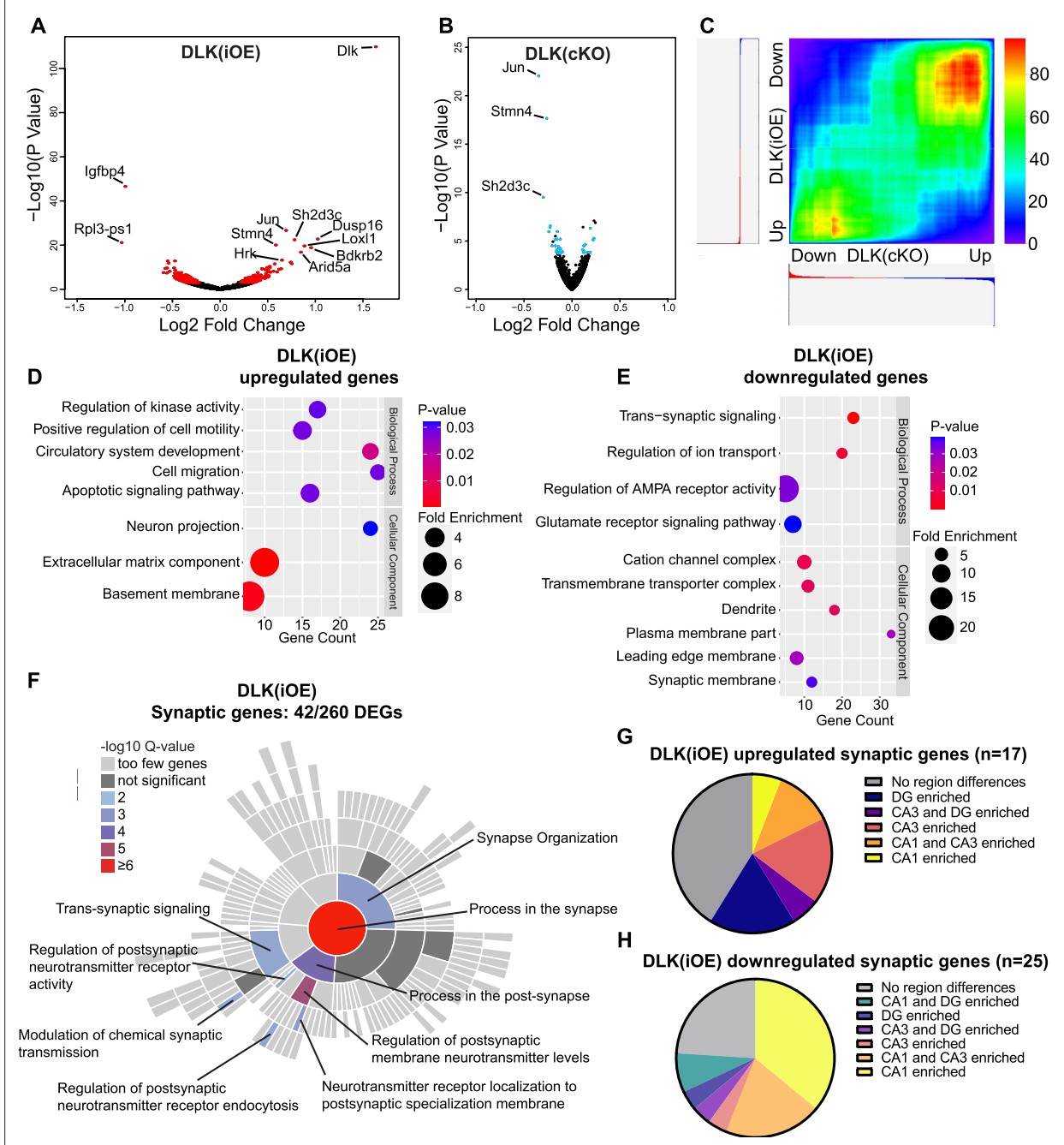

**Figure 3.** Differentially expressed genes revealed by RiboTag analysis of hippocampal glutamatergic neurons in DLK(cKO) and DLK(iOE) mice. (**A**) Volcano plot showing RiboTag analysis in *Slc17a7^Cre/+;*H11-DLK(iOE)/+;*Rpl22^HA/+* vs *Slc17a7^Cre/+;Rpl22^HA/+* (age P15). 260 genes (red) show differential expression with adjusted p-values <0.05 in *Slc17a7^Cre/+;*H11-DLK(iOE)/+, compared to control; names of genes with p<1E-10 are labeled. (**B**) Volcano plot showing RiboTag analysis in *Slc17a7^Cre/+;Map3k12^fl/fl;Rpl22^HA/+* vs *Slc17a7^Cre/+;Rpl22^HA/+* (age P15). 36 genes (blue) show differential expression with adjusted p-values <0.05; names of genes with p<1E-10 are labeled. (**C**) Rank-rank hypergeometric overlap (RRHO) comparison of gene expression in DLK(cKO) and DLK(iOE) RiboTag datasets shows enrichment of similar genes when DLK is low or high, respectively. Color represents the -log transformed hypergeometric p-values (blue for weaker p-value, red for stronger p-value). (**D, E**) Gene ontology (GO) analysis of significantly up- or down-regulated genes in hippocampal glutamatergic neurons of DLK(iOE) mice compared to the control. Colors correspond to p-values; circle size represents fold enrichment for the GO term; X position shows # of genes significantly enriched in the GO term. (**F**) SynGO sunburst plot shows enrichment of 42 differentially expressed genes from hippocampal glutamatergic neurons of DLK(iOE) mice, with color corresponding to significance. (**G, H**) Pie charts show distribution of the 42 synaptic genes up- or down- regulated in DLK(iOE), respectively, in CA1, CA3, and DG in dorsal hippocampus, based on in situ data (P56) in the Allen Mouse Brain Atlas.

*Figure 3 continued on next page*

*Figure 3 continued*

The online version of this article includes the following source data and figure supplement(s) for figure 3:

**Figure supplement 1.** Evidence for RiboTag immunoprecipitated samples and additional analysis of genes showing differential dependence on DLK expression levels.

**Figure supplement 1—source data 1.** Original western blots for images shown in *Figure 3—figure supplement 1A*.

**Figure supplement 1—source data 2.** PDF showing original western blot for image shown in *Figure 3—figure supplement 1A*, along with relevant band and genotype.

**Figure supplement 2.** Levels of c-Jun and p-c-Jun show dependency on expression levels of DLK.

genes both under normal conditions and when overexpressed, with a stronger effect on CA1 genes, compared to CA3 genes. Overall, the informatic analysis suggests that decreased expression of CA1 enriched genes may contribute to CA1 neuron vulnerability to elevated DLK, although it is also possible that the observed down-regulation of these genes is a secondary effect associated with CA1 neuron degeneration.

## DLK regulates translation of JUN and STMN4

The transcription factor c-Jun is a key downstream factor in DLK and JNK signaling (*Hirai et al., 2006*; *Itoh et al., 2009*; *Welsbie et al., 2017*). Our RiboTag analysis suggests that expression and translation of *Jun* mRNA to be significantly dependent on DLK expression levels (*Figure 3A and B*). To further test this observation, we performed immunostaining of hippocampal tissues using an antibody recognizing total c-Jun. In control mice, glutamatergic neurons in CA1 had low but detectable c-Jun immunostaining at P10 and P15, but reduced intensity at P60; those in CA3 showed an overall low level of c-Jun immunostaining at P10, P15, and P60; and those in DG showed a low level of c-Jun immunostaining at P10 and P15, and an increased intensity at P60 (*Figure 3—figure supplement 2A, C and E*). In DLK(iOE) mice at P10 when no discernable neuron degeneration was seen in any regions of hippocampus, only CA3 neurons showed a significant increase of immunostaining intensity of c-Jun, compared to control (*Figure 3—figure supplement 2A*). In P15 mice, we observed further increased immunostaining intensity of c-Jun in CA1, CA3, and DG, with the strongest increase (~four-fold) in CA1, compared to age-matched control mice (*Figure 3—figure supplement 2C*). The overall increased c-Jun staining is consistent with RiboTag analysis. We also analyzed DLK(cKO) mice at P60, and observed a trend for decreased c-Jun in CA3 (*Figure 3—figure supplement 2E*); the modest effects of DLK(cKO) on c-Jun proteins could be due to detection limitations for low levels of c-Jun. As phosphorylation of c-Jun (p-c-Jun) is known to reflect activation of DLK and JNK signaling (*Hirai et al., 2006*), we further investigated p-c-Jun levels in these mice. In control mice at P10, P15, only a few neuronal nuclei showed strong staining with p-c-Jun in CA1, CA3 and DG. In DLK(iOE) mice, we observed increased p-c-Jun-positive nuclei in CA1 at P10, and strong increase in CA1 (~tenfold), CA3 (~sixfold), and DG (~eightfold) at P15 (*Figure 3—figure supplement 2B and D*). The levels of p-c-Jun remained elevated in the surviving neurons in all three regions of DLK(iOE) mice at P60 (*Figure 2—figure supplement 2C*). In DLK(cKO) mice, p-c-Jun levels in CA3 showed a significant reduction, with the trend of reduced levels in CA1 and DG, compared to control mice (*Figure 3—figure supplement 2F*, P60). These results are consistent with a conclusion that translation of *Jun* mRNAs and phosphorylation of c-Jun show dependency on levels of DLK, with CA1 neurons showing higher dependence upon DLK overexpression.

The Stathmin family of proteins is thought to regulate microtubules through sequestering tubulin dimers (*Charbaut et al., 2001*; *Chauvin and Sobel, 2015*). This family of proteins includes four genes, all of which were identified in our hippocampal glutamatergic neuron translatome (*Figure 4—figure supplement 1A*), and only *Stmn4* showed significant up-regulation in DLK(iOE) and down-regulation in DLK(cKO), respectively (*Figure 3A and B*). We verified STMN4 protein expression by western blot analysis of hippocampal protein extracts. In control mice, we detected the levels of STMN4 to peak around P8 (*Figure 4—figure supplement 1B, C, F and G*). The abundance of STMN4 in DLK(cKO) and DLK(iOE) was subtly altered, which could be due to broad expression of STMN4 in hippocampus masking specific changes in glutamatergic neurons. We thus examined *Stmn4* mRNAs in hippocampus by RNAscope. *Stmn4* mRNAs were present in glutamatergic neurons across all regions of the hippocampus, with strongest expression in CA1 pyramidal neurons. While *Stmn4* mRNA puncta number in

these neurons was comparable between DLK(cKO) mice and control, in DLK(iOE) mice, glutamatergic neurons in CA1, CA3, and DG all showed upregulation of *Stmn4*, compared to control (*Figure 4A–C*, *Figure 4—figure supplement 2A, C*). These data support a role of DLK in modulating expression and translation of *Stmn4*.

## Elevated DLK signaling may disrupt microtubule homeostasis in hippocampal CA1 neurons

Substantial studies from other types of neurons in mice and invertebrate animals have linked DLK signaling with the regulation of microtubule cytoskeleton (*Asghari Adib et al., 2018*; *Jin and Zheng, 2019*; *Tedeschi and Bradke, 2013*). To assess whether DLK affects microtubules in hippocampal glutamatergic neurons, we performed Tuj1 immunostaining. We did not detect obvious changes in DLK(cKO) when compared to controls at P60 (*Figure 1E*). In DLK(iOE) mice at P15, expression levels and patterns of neuronal microtubules in each region of hippocampus appeared similar to control (*Figure 4D*, *Figure 4—figure supplement 2E*), although we found the overall Tuj1 staining pattern at P15 to be less defined and consistent. By P60, many CA1 neurons died and the hippocampus exhibited thinning of all strata within CA1, and the Tuj1 staining pattern became less organized in parallel dendrites in the stratum radiatum (SR) region of CA1 (*Figure 4I*). Increased Tuj1 staining in thin branches extended in varied directions, with bright staining seen in the apical dendrites near the pyramidal neuron cell body.

Several post-translational modifications of microtubules are thought to correlate with stable or dynamic state of microtubules. To explore whether DLK expression levels affected microtubule post-translational modifications, we performed immunostaining for acetylated tubulin, a modification generally associated with stable, longer-lived microtubules, and tyrosinated tubulin, a terminal amino acid that can be removed and is typically found on dynamic microtubules (*Janke and Magiera, 2020*). We detected no significant difference in the staining pattern and intensity of either tyrosinated tubulin or acetylated tubulin in DLK(cKO) mice, compared with age-matched control mice (*Figure 4N–R*). In DLK(iOE) mice at P15, both tubulin modifications showed no significant differences in pattern or intensity in CA1 SR, compared to age-matched control mice (*Figure 4E–H*), despite neuron death beginning in CA1. By P60, we observed increased staining intensity of acetylated tubulin and tyrosinated tubulin in the apical dendrites of surviving neurons in DLK(iOE) mice, particularly with tyrosinated tubulin staining revealing bright signals on small, thin branches (*Figure 4J–M*). To discern whether such microtubule modification changes were from neurons, we immunostained tissue sections with antibodies for MAP2, a neuron specific microtubule associated protein. We observed bright MAP2 signal in thin branches extending in varied directions in DLK(iOE) mice, compared to age-matched control mice (*Figure 4—figure supplement 2F and G*). Together this analysis suggests increased DLK expression may likely alter neuronal microtubule homeostasis and/or integrity.

## Increasing DLK expression alters synapses in dorsal CA1

A theme revealed in our hippocampal glutamatergic neuron RiboTag profiling suggests that translation of synaptic proteins may depend on the expression levels of DLK. To evaluate this observation, we examined synapses in the hippocampus by immunostaining for Bassoon, a core protein in the presynaptic active zone, Vesicular Glutamate Transporter 1 (VGLUT1) for synaptic vesicles, and Homer1, a post-synaptic scaffolding protein. In control mice and DLK(cKO) at P60, Bassoon staining in stratum radiatum (SR) of dorsal CA1, where CA3 neurons synapse onto CA1 dendrites, showed discrete puncta that were mostly apposed to the postsynaptic marker Homer1, representing properly formed synapses (*Figure 5A*). We measured size and density by counting Bassoon and Homer1 puncta and the sites where Bassoon and Homer1 overlap, a proxy for synapses. We detected no significant difference in DLK(cKO), compared to control (*Figure 5A–F*). To assess effects of DLK overexpression on synapses, we immunostained hippocampal sections from both P10 and P15, with age-matched littermate controls. Quantification of Bassoon and Homer1 immunostaining revealed no significant differences in CA1 SR and CA3 SR and SL in P10 mice of DLK(iOE) and control (*Figure 5—figure supplement 2A–F*, *Figure 5—figure supplement 3A–J*). In P15, Bassoon density and size in CA1 SR were comparable in both mice (*Figure 5G, H and K*), while Homer1 density and size were reduced in DLK(iOE) (*Figure 5G, I and L*). Overall synapse number in CA1 SR was similar in DLK(iOE) and control mice (*Figure 5J*). Similar analysis on CA3 SR and SL detected no significant difference from control

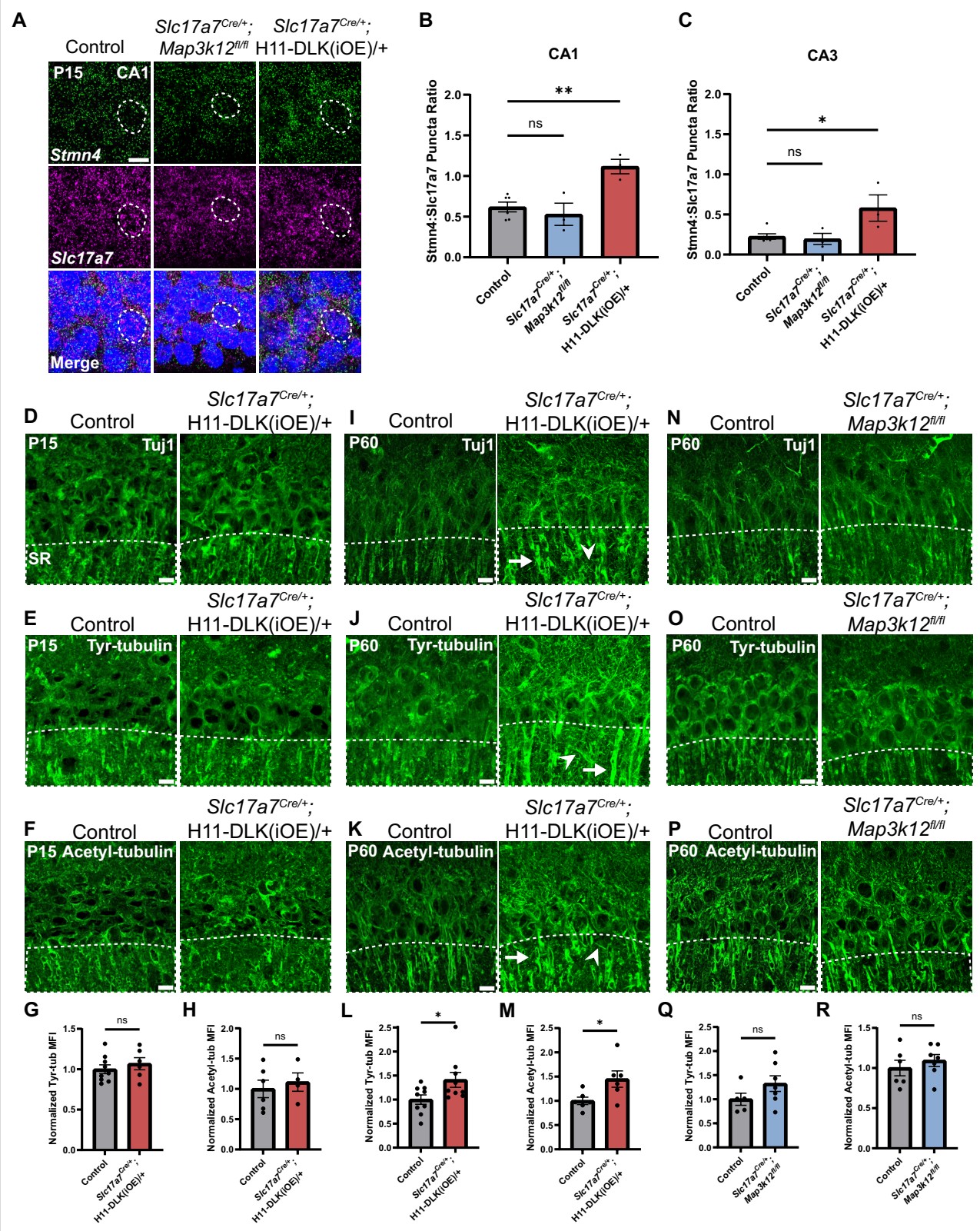

**Figure 4.** *Stmn4* and microtubule homeostasis show dependency on the expression levels of DLK. (**A**) Confocal single-slice image of RNAscope analysis of *Stmn4* and *Slc17a7* mRNAs in hippocampal neurons. Dashed circle outlines single nuclei. Scale bar, 10 μm. (**B, C**) Quantification of the ratio of *Stmn4* to *Slc17a7* RNAscope puncta in same nuclei of CA1 and CA3 neurons, respectively. N=6,3,3 mice of respective genotypes, quantified from >50 cells per genotype from 4 sections per mouse. Statistics: One way ANOVA with Dunnett's multiple comparison test, ns, not significant; * p<0.05; ** p<0.01.

*Figure 4 continued on next page*

*Figure 4 continued*

(**D–F**) Confocal z-stack (max projection) images of CA1 immunostained for Tuj1, tyrosinated tubulin, and acetylated tubulin, respectively, in control and *Slc17a7$^{Cre/+}$*·H11-DLK(iOE)/+ mice of P15. SR: stratum radiatum. (**G, H**) Normalized mean fluorescence intensity (MFI) of tyrosinated and acetylated tubulin, respectively, after thresholding signals in SR in CA1 (dashed outlines on images in E-F). N=9, 6 mice, 3 sections averaged per mouse in (**G**); N=6, 4 mice, 3 sections averaged per mouse in (**H**). (**I–K**) Confocal z-stack (max projection) images of immunostained CA1 sections for Tuj1, tyrosinated tubulin, and acetylated tubulin, respectively, in control and *Slc17a7$^{Cre/+}$*·H11-DLK(iOE)/+ mice of P60. (**L, M**) Normalized MFI of tyrosinated and acetylated tubulin, respectively, after thresholding signal in SR in CA1 (dashed outlines on images in J-K). N=9, 9 mice, 3 sections averaged per mouse in L; N=6, 6 mice, 3 sections averaged per mouse in M. (**N–P**) Confocal z-stack (max projection) images of immunostained CA1 sections for Tuj1, tyrosinated tubulin, and acetylated tubulin, respectively, in control and *Slc17a7$^{Cre/+}$*;*Map3k12$^{fl/fl}$* mice of P60. (**Q, R**) Normalized MFI for tyrosinated and acetylated tubulin, respectively, after thresholding signal in SR in CA1 (dashed outlines on images in O-P). N=5, 7 mice, 3 sections averaged per mouse in Q; N=6, 7 mice, 3 sections averaged per mouse in R. All tubulin images shown as maximum projection of z-stack. Scale bar, 10 μm. In I-J, arrows point to apical dendrites with elevated immunostaining signal; arrowheads point to thin neurites with elevated signal. Statistics in (**G, H, L, M, Q, R**): Unpaired t-test. ns, not significant; * p<0.05. All error bars represent SEM.

The online version of this article includes the following source data and figure supplement(s) for figure 4:

**Figure supplement 1.** Stathmin transcript abundance and western blot analysis of DLK, STMN4 in mice aged P10 to 1 yr.

**Figure supplement 1—source data 1.** Original western blots for images shown in *Figure 4—figure supplement 1B and C*.

**Figure supplement 1—source data 2.** PDF showing original western blots for images shown in *Figure 4—figure supplement 1B and C*, along with relevant bands and genotypes.

**Figure supplement 2.** Additional evidence for *Stmn4* and microtubule expression in DLK(cKO) and DLK(iOE) in hippocampus.

(*Figure 5—figure supplement 3M–V*). Staining of VGLUT1 protein showed less discrete puncta than those of Bassoon or Homer1, with small puncta and larger clusters of puncta close together (*Figure 5—figure supplement 1A and D*). In DLK(cKO) we observed a trend towards an increased number of VGLUT1 puncta (p=0.0653) with no change to puncta size (*Figure 5—figure supplement 1A–C*). In DLK(iOE) we observed fewer VGLUT1 puncta in SR, consistent with the analysis on Homer1 at P15, with no significant change to puncta size (*Figure 5—figure supplement 1D–F*). These data reveal that while conditional knockout of DLK may not have a strong effect on glutamatergic synapses, increased expression of DLK leads to mild alteration in the CA1 region at P15, correlating with the onset of CA1 neurodegeneration.

## High levels of DLK cause short neurite formation in primary hippocampal neurons

To gain better resolution on how DLK expression levels affect glutamatergic neuron morphology and synapses, we next turned to primary hippocampal cultures. To enable visualization of Slc17a7-positive neurons, we introduced a floxed Rosa26-tdTomato reporter (*Madisen et al., 2010*) into *Slc17a7$^{Cre/+}$*, *Slc17a7$^{Cre/+}$*;*Map3k12$^{fl/fl}$*, and *Slc17a7$^{Cre/+}$*;H11-DLK(iOE)/+ mice. We prepared primary hippocampal neurons from P1 pups of respective crosses (Materials and methods), so around ¼ of glutamatergic (Slc17a7-Cre) neurons in the cultures had both tdTomato and the genotype of interest (*Map3k12$^{fl/fl}$*, WT, or H11-DLK(iOE)/+). We did not notice an obvious effect of DLK(iOE) or DLK(cKO) on neuron density in cultures at DIV2. To assess neuronal type distribution in our cultures, we immunostained DIV14 neurons with antibodies for Satb2, as a CA1 marker (*Nielsen et al., 2010*), and Prox1, as a marker of DG neurons (*Iwano et al., 2012*). We did not observe significant differences in the proportion of cells labeled with each marker in DLK(cKO) or DLK(iOE) cultures (*Figure 6—figure supplement 1E*). These data are consistent with the idea that DLK signaling does not have a strong role in neuron-type specification both in vivo and in vitro.

We verified DLK protein pattern and levels by immunostaining with DLK antibodies (*Figure 6A*). In DIV2 neurons from control mice, DLK was present in cell soma, likely reflecting Golgi apparatus localization as reported (*Hirai et al., 2002*), and showed a punctate pattern in neurites, particularly the axon growth cone regions. We also immunostained for STMN4 and observed a similar punctate localization in the cell soma, neurites, and growth cones (*Figure 6A*), in line with published data (*Chauvin et al., 2008*; *Gavet et al., 2002*). STMN4 puncta appeared to be non-overlapping with DLK (*Figure 6—figure supplement 1B*). In DIV2 neurons from DLK(cKO), STMN4 exhibited a similar punctate pattern, with intensity comparable to that in control neurons. In neurons from DLK(iOE), DLK levels were increased, and STMN4 levels were also increased (Spearman correlation r=0.7454) (*Figure 6A and B*), supporting our RiboTag analysis. Expression of another member of

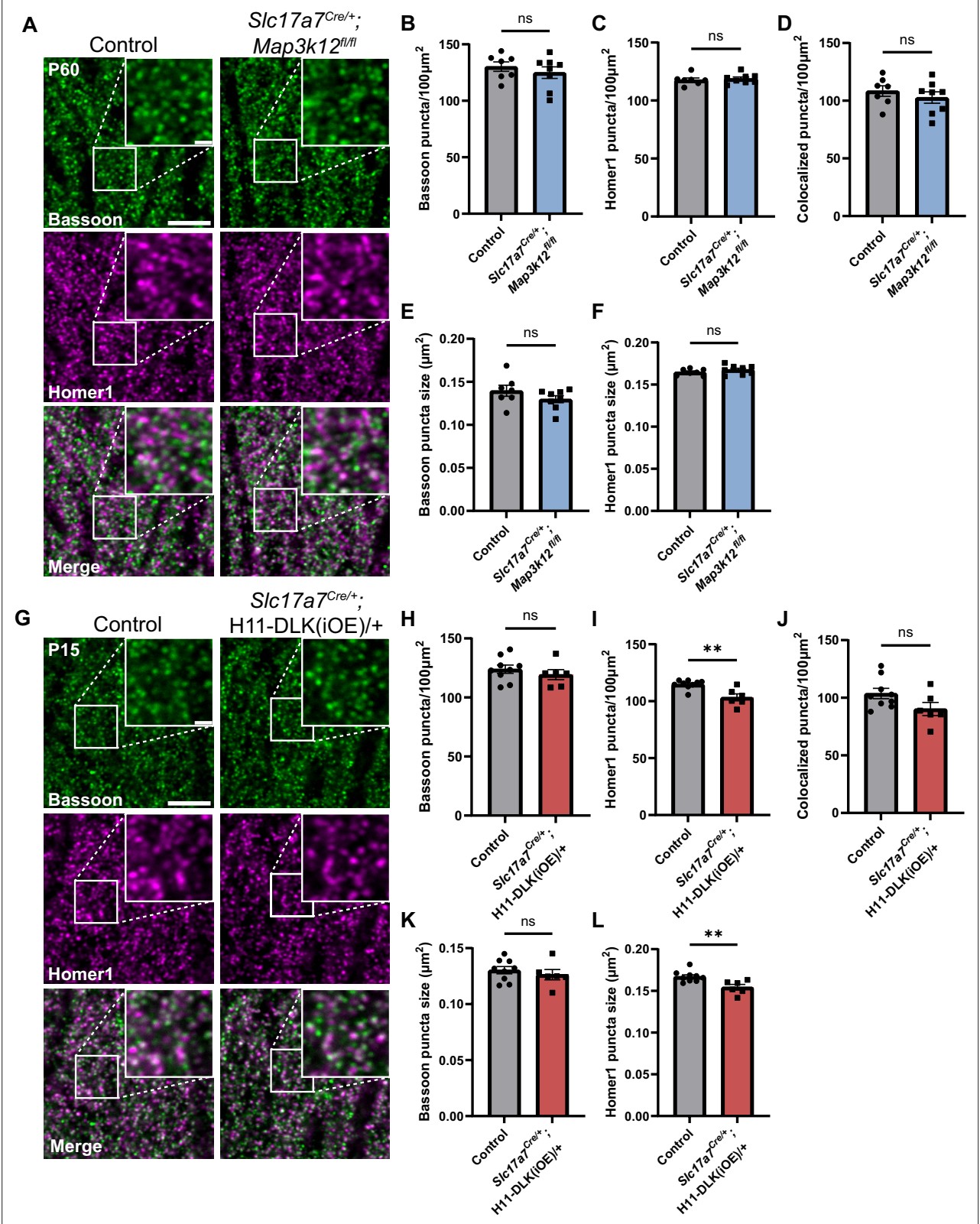

**Figure 5.** Hippocampal dorsal CA1 glutamatergic neurons show altered synapses following increased DLK expression. (**A**) Confocal single-slice images of Bassoon and Homer1 immunostaining in CA1 stratum radiatum (SR) of control and *Slc17a7^{Cre/+};Map3k12^{fl/fl}* mice of P60. (**B, C**) Quantification of Bassoon and Homer1 puncta density, respectively. (**D**) Quantification of co-localization of Bassoon and Homer1. (**E, F**) Quantification of Bassoon and Homer1 puncta size. Data points represent average values per mouse from 3 sections. N=7 control, and 8 *Slc17a7^{Cre/+};Map3k12^{fl/fl}* mice. Statistics:

*Figure 5 continued on next page*

*Figure 5 continued*

unpaired t-test or Mann-Whitney U test if not passing normality. ns, not significant. (**G**) Confocal single-slice images of Bassoon and Homer1 immunostaining in CA1 SR of control and *Slc17a7Cre/+*;H11-DLK(iOE)/+ mice of P15. (**H–I**) Quantification of Bassoon and Homer1 puncta density, respectively. (**J**) Quantification of co-localization of Bassoon and Homer1. (**K, L**) Quantification of Bassoon and Homer1 puncta size. Data points represent average values per mouse from 3 sections, N=9 control, and 6 *Slc17a7Cre/+*;H11-DLK(iOE)/+ mice. Statistics: unpaired t-test or Mann-Whitney U test if not passing normality. ns, not significant; ** p<0.01. Scale bars, 5 µm in panel images, and 1 µm in enlarged images. All error bars represent SEM.

The online version of this article includes the following figure supplement(s) for figure 5:

**Figure supplement 1.** VGLUT1 pattern in dorsal CA1 in DLK(cKO) and DLK(iOE).

**Figure supplement 2.** Analysis of Bassoon and Homer1 immunostaining in dorsal CA1 in P10 of DLK(iOE).

**Figure supplement 3.** Analysis of Bassoon and Homer1 immunostaining in CA3 synapses of DLK(iOE) at P10 and P15.

Stathmin, STMN2, is associated with DLK-dependent responses in DRG neurons (*Summers et al., 2020*; *Thornburg-Suresh et al., 2023*). Although our RiboTag analysis did not identify significant changes of *Stmn2* (*Figure 4—figure supplement 1A*), we tested whether STMN2 protein levels could be altered in our cultured hippocampal neurons. In DIV2 control neurons STMN2 staining showed punctate localization in the perinuclear region (*Gavet et al., 2002*; *Lutjens et al., 2000*), along with punctate signals in neurites and growth cones, similar to STMN4. By co-immunostaining analysis of DLK and STMN2, we detected a positive correlation between DLK and STMN2 (*Figure 6—figure supplement 1C and D*, Spearman correlation $r$=0.4693), albeit to a moderate level in comparison to that of DLK and STMN4. These data suggest that in hippocampal glutamatergic neurons DLK has a stronger effect on STMN4 levels but may also regulate protein levels of STMN2.

In hippocampal cultures at DIV2 neurites are actively growing and establish thicker branches, which form dendrites and axons (*Dotti et al., 1988*). In our control cultures at DIV2, the majority of tdTomato labeled neurons developed multiple neurites from the cell soma, with one neurite developing into an axon, defined here as a neurite longer than 90 µm (*Figure 6—figure supplement 1A*). Additionally, thin, often short, neurites were observed branching off from the cell soma, axons, dendrites, and growth cones. In DLK(cKO) cultures at DIV2, we observed a trend of more neurons without an axon (*Figure 6C*), though the differentiated axons appeared morphologically indistinguishable from control. The total number of neurites around the cell soma in DLK(cKO) neurons was significantly reduced, compared to control (*Figure 6A and D*). In DLK(iOE) cultures at DIV2, we observed a significant increase in the percentage of neurons without an axon and also neurons with multiple axons, compared to control cultures (*Figure 6C*, *Figure 6—figure supplement 1A*). Moreover, neurons expressing high levels of DLK protein displayed an increased number of neurites either around the cell soma as primary neurites or as secondary neurites, compared to control (*Figure 6A and D*). Such neurites were typically thin, and frequently appeared short, like filopodia. These thin neurites sometimes developed a rounded tip and showed beading appearance, resembling degeneration (*Figure 6A and E*). These data suggest a role for DLK in neurite formation and axon specification in cultured hippocampal glutamatergic neurons.

We further analyzed microtubules in individual neurites of the DIV2 neurons. Control neurons exhibited staining for tyrosinated tubulin in differentiated axons and dendrites as well as filopodia and towards peripheral regions of growth cones (*Figure 6E*). Acetylated tubulin was present in differentiated axons and dendrites and in the central region of growth cones where stable microtubules were present (*Figure 6F*), and was absent from filopodia and microtubules in the peripheral regions of growth cones. The thin neurites in neurons expressing very high levels of DLK appeared to have thin bundles of microtubules, and these neurites generally were not associated with a growth cone or microtubules splaying apart at the end as was common in WT growth cones (*Figure 6E*). Neurites from neurons with high DLK expression also had tyrosinated tubulin, while acetylated tubulin was frequently absent (*Figure 6E and F*). Additionally, STMN4 was present in the thin neurites, especially in those with high levels of DLK (*Figure 6A*), suggesting the thin neurites may likely be dynamic in nature. These results suggest that in cultured hippocampal neurons high levels of DLK promotes formation of short, thin, dynamic branches.

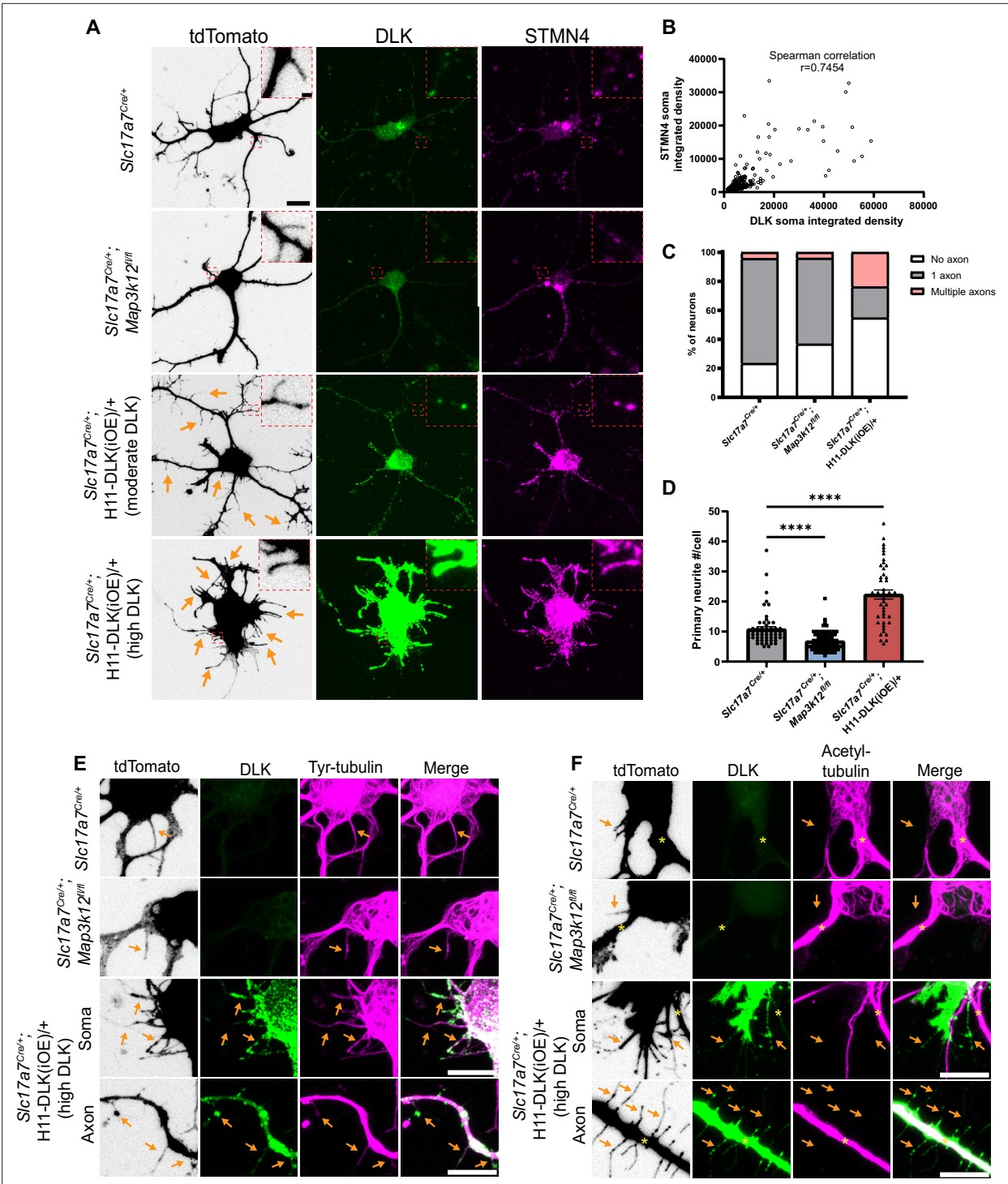

**Figure 6.** DLK promotes short neurite formation in primary cultured hippocampal neurons. (**A**) Confocal images of DIV2 primary hippocampal glutamatergic neurons immunostained with DLK and STMN4. Neurons with indicated genotypes are labeled by tdTomato from Cre-dependent Rosa26-tdTomato, generated from hippocampi in P1 pups from the following crosses: for control: *Slc17a7*$^{Cre/+}$ X *Rosa26*$^{tdT/+}$; for DLK(cKO): *Slc17a7*$^{Cre/+}$;*Map3k12*$^{fl/fl}$ X *Map3k12*$^{fl/fl}$;*Rosa26*$^{tdT/+}$; for DLK(iOE): H11-DLK(iOE)/H11-DLK(iOE) X *Slc17a7* $^{Cre/+}$;*Rosa26*$^{tdT/+}$. Orange arrows point to some of the thin neurites from neurons overexpressing DLK. Red dashes outline enlarged view of neurites. Scale bar, 10 µm neuron, 1 µm enlarged view. (**B**) Graph shows positive correlation between STMN4 immunostaining, measured as integrated density (Area X MFI) in neuronal soma, to integrated density of DLK immunostaining. N≥3 cultures/genotype,≥60 cells/genotype. Spearman correlation *r*=0.7454. (**C**) Quantification of percentage of neurons with no, one, or more than one axon (defined by neurites longer than 90 µm) in each genotype. Number of neurons: 47 from 3 Slc17a7$^{Cre}$ (control) cultures, 49 from 3 DLK(cKO) cultures, 42 from 4 DLK(iOE) cultures. Statistics: Fisher's exact test shows significance (p<0.0001) between genotype and number of axons.

*Figure 6 continued on next page*

*Figure 6 continued*

Pairwise comparisons with Fisher's exact test: Axon formation in control vs DLK(cKO): p=0.1857. Formation of multiple axons in control vs DLK(cKO): p>0.9999. Axon formation in control vs DLK(iOE): p=0.0042. Formation of multiple axons in control vs DLK(iOE): p=0.0001. (**D**) Quantification of number of primary neurites, which include both branches and filopodia, per neuron. Number of neurons: 55 from 4 Slc17a7$^{Cre}$ (control) cultures, 70 from 4 DLK(cKO) cultures, 45 from 5 DLK(iOE) cultures. Statistics, Kruskal-Wallis test with Dunn's multiple comparison test. **** p<0.0001. Error bars represent SEM. (**E**) Confocal z-stack images of tyrosinated tubulin immunostaining from DIV2 cultures of genotypes indicated, showing that filopodia structures (arrows) around the soma and axons of neurons with high expression of DLK have tyrosinated tubulin. (**F**) Confocal z-stack images of acetylated tubulin immunostaining from DIV2 cultures of genotypes indicated, showing that filopodia structures (arrows) around the soma and axons of neurons with high expression of DLK do not have acetylated tubulin. Asterisks indicate stable branches containing acetylated tubulin. Scale bar in E, F, 10 μm. Tyrosinated tubulin and acetylated tubulin staining shows saturated appearance to visualize staining in thin neurites.

The online version of this article includes the following source data and figure supplement(s) for figure 6:

**Figure supplement 1.** Analysis of Stathmins in primary cultured hippocampal neurons from DLK(cKO) and DLK(iOE).

**Figure supplement 2.** Comparison of STMN2 and STMN4 antibodies.

**Figure supplement 2—source data 1.** Original western blots for images shown in *Figure 6—figure supplement 2B*.

**Figure supplement 2—source data 2.** PDF showing original western blots for images shown in *Figure 6—figure supplement 2B*, along with relevant bands and genotypes.

## Increased DLK expression alters synapses in primary hippocampal neurons

Our RiboTag data showed enrichment of synaptic genes in both DLK(cKO) and DLK(iOE) (*Figure 3F*, *Figure 3—figure supplement 1J*, *Supplementary file 1* WT vs DLK(cKO) DEGs, *Supplementary file 2* WT vs DLK(iOE) DEGs). Some of these genes function in cell adhesion, calcium signaling, and AMPA receptor expression, which may affect dendritic spine morphology and synaptic connections. Increased DLK levels led to reduced Homer1 density in hippocampal tissue (*Figure 5I*). To further investigate the effects of DLK on synapses, we immunostained the cultured hippocampal neurons at DIV14 with Bassoon. The control neurons showed discrete Bassoon puncta in axons (*Figure 7A–C*). DLK(cKO) neurons showed no significant change in Bassoon puncta size or density. In contrast, DLK(iOE) neurons showed abnormal Bassoon staining that was larger and irregular in shape (*Figure 7A–C*), suggesting that high levels of DLK disrupted presynaptic active zones.

Morphology of dendritic spines is associated with differences in maturity, with mushroom spines representing a more mature morphology than thin spines (*Yoshihara et al., 2009*). To assess how DLK expression levels affect dendritic spine morphology and frequency in cultured neurons, we evaluated both the density and type of dendritic spines formed at DIV14 on neurons with spines, visualized by tdTomato. We categorized spine morphology in different types, following previous studies (*Risher et al., 2014*). Neurons from DLK(cKO) showed spines at a higher density than control neurons, with significantly more mushroom spines (*Figure 7D–G*). In contrast, neurons from DLK(iOE) cultures had a reduced density of dendritic spines compared to control neurons (*Figure 7D and E*). DLK(iOE) neurons also formed spines that tended to be more immature, with significantly fewer mushroom spines and a higher percentage of thin spines (*Figure 7D, F and G*). These results reveal that expression levels of DLK appear to be inversely correlated with spine density and maturity.

## Discussion

### Selective vulnerability of hippocampal glutamatergic neurons to increased DLK expression

Under normal conditions, the abundance of endogenous DLK in many parts of the brain is generally kept at a low level. Elevated DLK signaling has been associated with traumatic injury and implicated in Alzheimer's disease and other neurodegenerative conditions (*Asghari Adib et al., 2018*; *Huang et al., 2017*; *Jin and Zheng, 2019*; *Le Pichon et al., 2017*; *Tedeschi and Bradke, 2013*). Despite its broad expression, we know little about DLK's role in the central nervous system. In this study, we combined conditional knockout and overexpression of DLK to uncover its roles in the hippocampal glutamatergic neurons. Our finding that conditional deletion of DLK in the glutamatergic neurons using *Slc17a7$^{Cre}$* in late embryonic development does not cause discernable morphological defects is consistent with the previous reports that hippocampal neurons are largely normal in constitutive

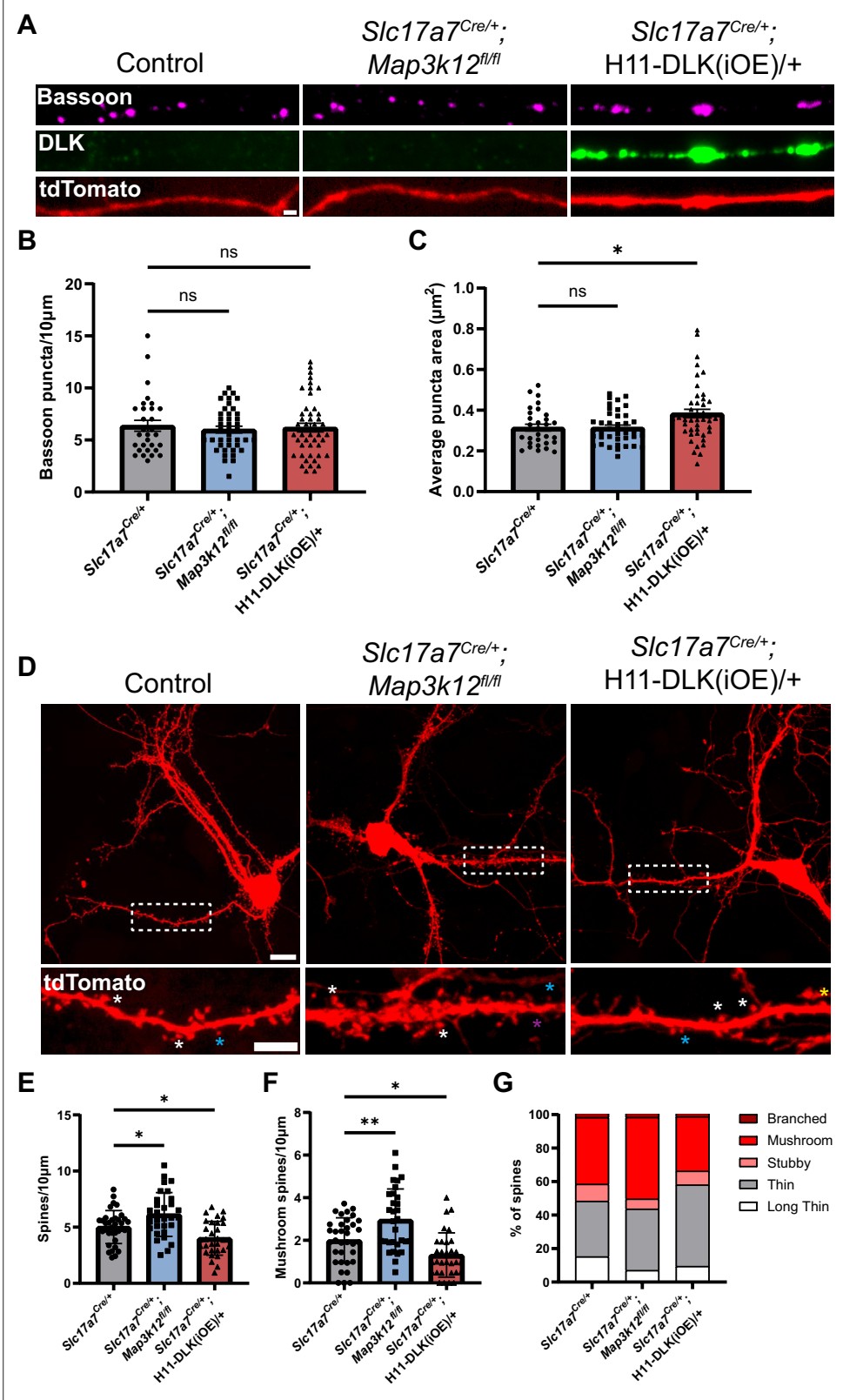

**Figure 7.** Increasing DLK expression alters synapse formation in primary cultured hippocampal neurons. (**A**) Confocal images of axons of DIV14 neurons of indicated genotype, co-stained with Bassoon and DLK. Neurons with indicated genotypes are labeled by tdTomato from Cre-dependent Rosa26-tdTomato generated from the following crosses: for control: *Slc17a7^Cre/+* X *Rosa26^tdT/+*; for DLK(cKO): *Slc17a7^Cre/+;Map3k12^fl/fl* X

*Figure 7 continued on next page*

*Figure 7 continued*

*Map3k12^{fl/fl};Rosa26^{tdT/+}*; for DLK(iOE): H11-DLK(iOE)/H11-DLK(iOE) X *Slc17a7 ^{Cre/+};Rosa26^{tdT/+}*. Scale bar, 1 μm. (**B**) Quantification of bassoon puncta density. (**C**) Quantification of average bassoon puncta size from individual neurons. Number of neurons: 30 from 3 Slc17a7-cre (control) cultures, 41 from 3 DLK(cKO) cultures, 46 from 4 DLK(iOE) cultures. Statistics: One way ANOVA with Dunnett's multiple comparison test. ns, not significant; * p<0.05. (**D**) Confocal z-stack images of DIV14 neurons of indicated genotype, labeled by Rosa26-tdTomato. Dashed boxes outline dendrites enlarged below for dendritic spines. Asterisks provide some examples of spine types; long thin with purple; thin with blue; mushroom with white; stubby with yellow. Scale bar, 10 μm top, 5 μm bottom. (**E**) Quantification of dendritic spine density. (**F**) Quantification of mushroom spine density. (**G**) Distribution of spine types. (**E–G**) Number of neurons: 35 from 3 Slc17a7-cre (control) cultures, 31 from 3 DLK(cKO) cultures, 31 from 3 DLK(iOE) cultures. Statistics: One way ANOVA with Dunnett's multiple comparison test. * p<0.05; ** p<0.01. All error bars represent SEM.

knockout of DLK (*Hirai et al., 2006*; *Hirai et al., 2011*). In contrast, induced overexpression of DLK, which leads to activation of JNK signaling evidenced by increased p-c-Jun, causes the glutamatergic neurons in dorsal CA1 and dentate gyrus to undergo pronounced death, while CA3 neurons appear less vulnerable even under chronic elevated DLK expression. The levels of DLK in our DLK(iOE) mice model appear comparable to those reported under traumatic injury and chronic stress. The pattern of DLK-induced neuronal death shares similarity to the differential vulnerability of CA1 and CA3 neurons reported in patients with Alzheimer's disease (*West et al., 1994*), and animal models of oxidative stress (*Wilde et al., 1997*), ischemia (*Smith et al., 1984*), and glutamate excitotoxicity from NMDA (*Vornov et al., 1991*). The dorsal-ventral hippocampal neuron death pattern associated with increased expression of DLK is also similar to that observed in animal models of ischemia (*Smith et al., 1984*). Such regional differences of hippocampal neurons in response to insults or genetic manipulation may be attributed to multiple factors, such as the nature of the neural network (*Viana da Silva et al., 2024*), intrinsic differences between CA1 and CA3 neurons in their abilities to buffer calcium changes, mitochondrial stress, protein homeostasis, glutamate receptor distribution (*Schmidt-Kastner, 2015*), and as discussed further, the degree to which transcription factors, such as p-c-Jun or other AP1 factors, are activated under different conditions.

## DLK-dependent cellular network exhibits commonality and cell-type specificity

DLK to JNK signaling is known to lead to transcriptional regulation. Several studies have used transcriptomic profiling to reveal DLK-dependent gene expression in different regions of the brain, such as cerebellum and forebrain, and in specific neuron types, such as DRG neurons and RGC neurons following axon injury or nerve growth factor withdrawal (*Goodwani et al., 2020*; *Hu et al., 2019*; *Larhammar et al., 2017*; *Le Pichon et al., 2017*; *Shin et al., 2019*; *Watkins et al., 2013*). One recent study reported RiboTag profiling of the DLK-dependent gene network in axotomized spinal cord motor neurons (*Asghari Adib et al., 2024*). In agreement with the overall findings from these studies, we find that loss of DLK in hippocampal glutamatergic neurons results in modest expression changes in a small number of genes, while overexpression of DLK leads to expression changes in a larger set of genes. Gene ontology analysis of our hippocampal glutamatergic neuron translatome reveals a similar set of terms as found in the other expression studies, including neuron differentiation, apoptosis, ion transport, and synaptic regulation.

Comparison of the translational targets of DLK in our study with these prior analyses also shows notable differences that are likely specific to neuron-type and contexts of experimental manipulations. For example, we find a strong induction of *Jun* translation associated with increased expression of DLK, but no significant changes in *Atf3* or *Atf4* translation, which were reported to show DLK-dependent increases in axotomized spinal cord motoneurons, and injured RGCs and DRGs (*Asghari Adib et al., 2024*; *Larhammar et al., 2017*; *Shin et al., 2019*; *Watkins et al., 2013*). Most ATF4 target genes (*Somasundaram et al., 2023*) also show no significant changes in our hippocampal glutamatergic neuron translatome. Moreover, we find a cohort of synaptic genes showing expression dependency on DLK (such as *Tenm3, Nptx1*, and *Nptxr),* but not any of the complement genes (*C1qa, C1qb, C1qc*), which are up-regulated in the regenerating spinal cord motor neurons where neuron-immune cell interaction has a critical role (*Asghari Adib et al., 2024*). Genomic structure and

regulation intrinsic to the cell type may be a major factor underlying the gene expression differences in ours and other studies. Elevated DLK signaling in axotomized neurons may promote a strong regenerative response through activation of transcription factors, such as ATF3 and ATF4, whereas JUN and others that are actively expressed in hippocampal neurons may lead to a strong effect on refining synapses in response to DLK signaling. Overall, ours and the previous studies underscore the importance of systematic dissection of molecular pathways to understand neuron-type specific functionality to DLK signaling.

Our analysis of the spatial expression patterns of genes that showed association with DLK expression levels provides molecular insight to the differential vulnerability of hippocampal glutamatergic neurons under neurodegenerative conditions. We find that a select set of genes enriched in CA1 are up-regulated in DLK knockout and down-regulated upon DLK overexpression. The c-Jun transcription factor has a key role in hippocampal cell death responses as mutations preventing c-Jun phosphorylation led to decreased neuronal apoptosis in the hippocampus following treatment with kainic acid (*Behrens et al., 1999*). Basal levels of c-Jun and phosphorylated c-Jun in hippocampus are generally low (*Goodwani et al., 2020*; *Pozniak et al., 2013*). We find modest reductions in p-c-Jun in DLK(cKO) glutamatergic neurons, consistent with previous studies of the constitutive knockout of DLK (*Hirai et al., 2006*). In contrast, in DLK(iOE) neurons, translation of c-Jun and phosphorylation of c-Jun are increased, with CA1 neurons exhibiting higher increase than CA3 neurons. The c-Jun promoter has consensus AP1 sites, and c-Jun can regulate its own expression levels in cancer cell lines (*Angel et al., 1988*), NGF-deprived sympathetic neurons (*Eilers et al., 1998*) and kainic acid treated hippocampus (*Mielke et al., 1999*). While our data does not pinpoint the molecular changes explaining why CA3 would show less vulnerability to increased DLK, we may speculate that DLK(iOE) induced signal transduction amplification may differ in CA1 vs CA3. CA1 genes appear to be more strongly regulated than CA3 genes, consistent with our observation that increased c-Jun expression in CA1 is greater than that in CA3. Other parallel molecular factors may also contribute to resilience of CA3 neurons to DLK(iOE), such as HSP70 chaperones, different JNK isoforms, and phosphatases, some of which showed differential expression in our RiboTag analysis of DLK(iOE) vs WT (*Supplementary file 2* WT vs DLK(iOE) DEGs). Together with other genes that show dependency on DLK, the DLK and Jun regulatory network contributes to the regional differences in hippocampal neuronal vulnerability under pathological conditions.

## Conserved functions of DLK in regulating Stathmins

Stathmins are tubulin binding proteins broadly expressed in many types of neurons. Several studies have reported that DLK can regulate the expression of different Stathmin isoforms in multiple neuron types under injury conditions (*Asghari Adib et al., 2024*; *DeVault et al., 2024*; *Hu et al., 2019*; *Larhammar et al., 2017*; *Le Pichon et al., 2017*; *Shin et al., 2019*). In the hippocampus *Stmn2* is expressed at a higher level than *Stmn4*, with the relative ratios of *Stmn4:Stmn2* in hippocampus much higher than in DRGs (*Zeisel et al., 2018*). We find that DLK can modulate expression and translation of *Stmn4* in hippocampal neurons. ChIP-seq data for Jun from ENCODE (ENCSR000ERO) suggest a possible binding site in the promoter region of Stmn4 (*The ENCODE Project Consortium, 2012*; *Luo et al., 2020*). STMN4 expression in hippocampus peaks around P8, correlating to neurite outgrowth and synapse formation and pruning (*Paolicelli et al., 2011*). At the level of hippocampal tissue, loss of DLK causes no detectable changes to microtubules, while increased levels of DLK appear to alter microtubule homeostasis in dendrites, with generally increased levels of both stable and dynamic microtubule markers. The CA1 neurons in DLK(iOE) also show fewer parallel microtubule arrays of apical dendrites, with short branches extending in varied directions. Our results from primary hippocampal neurons support roles for DLK in both short neurite and axon formation, similar to observations in cortical neurons, where DLK contributes to stage specific regulation of microtubules (*Hirai et al., 2011*). In primary cortical neurons overexpression of STMN4 can increase neurite length and branching when an epigenetic cofactor regulating MT dynamics was knocked down (*Tapias et al., 2021*). We speculate that DLK-dependent regulation of STMN4 and other STMNs may have a critical role in the long-term cytoskeletal rearrangements for neuronal morphology and synapse formation or stability. Nonetheless, as Stmns have considerable redundancy in expression and function, changes in STMN4 alone are unlikely to be a major factor for the observed hippocampal regional neuron death.

## Conserved roles of DLK in synapse formation and maintenance

The in vivo functions of the DLK family of proteins were first revealed in studies of synapse formation in *C. elegans* and *Drosophila* (*Collins et al., 2006*; *Nakata et al., 2005*). Our hippocampal glutamatergic neuron translatomic data extends this function by revealing a strong theme of DLK-dependent network in synapse organization, adhesion molecules and regulation of trans-synaptic signaling, especially related to AMPA receptor expression and calcium signaling, such as Neuronal Pentraxin 1 and Neuronal Pentraxin Receptor *Nptx1* and *Nptxr* (*Gómez de San José et al., 2022*). From our synapse analysis in culture, we find increased DLK alters pattern of presynaptic protein Bassoon, consistent with the findings on *C. elegans* and *Drosophila* synapses (*Nakata et al., 2005*; *Collins et al., 2006*). We also find DLK regulates dendritic spine morphology, with loss of DLK associated with a greater number of spines with more mature spine morphology, while increased DLK was associated with fewer and less mature spines. These results are similar to that observed in layer 2/3 cortical neurons where loss of DLK is associated with larger dendritic spines in developing neurons and higher density of spines when exposed to Aβ plaques, which lead to loss of nearby spines (*Le Pichon et al., 2017*; *Pozniak et al., 2013*). In CA1 dendritic regions, DLK overexpression reduced Homer1 density, suggesting synaptic defects may correlate with the onset of degeneration. In axotomized spinal cord motor neurons DLK induces activation of complement, leading to microglial pruning of synapses in injured motoneurons (*Asghari Adib et al., 2024*). These data support a conserved role of DLK in synapse formation and maintenance, through regulating the translation of genes involved in neuron outgrowth, synaptic adhesion, and synapse activity.

## Limitation of our study

We have investigated roles of DLK in hippocampal glutamatergic neuron development, synapse regulation, and neuron death processes. We infer that DLK-dependent expression and translation of CA1 enriched genes may likely play roles in regional vulnerability to increased DLK signaling. However, our RiboTag profiling was performed with whole hippocampus at time when CA1 death was noticeable. Our analysis of spatial expression patterns of DLK-dependent genes relies on available data from adult animals, which may not reflect the patterns at P15, or in response to altered DLK. We cannot rule out that some of the decreased expression of CA1 enriched genes in DLK(iOE) could be secondary due to neuronal death that could result in fewer CA1 neurons present in our mRNA samples. Our analysis also does not directly address why CA3 neurons are less vulnerable to increased DLK expression. Future studies using cell-type specific RiboTag profiling and other methods at a refined time window will be required to address how DLK-dependent signaling interacts with other networks underlying hippocampal regional neuron vulnerability to pathological insults. While we find evidence for apoptosis, other forms of cell death may also occur. Additional experiments will be needed to elucidate in vivo roles of STMN4 and its interaction with other STMNs. It is worth noting that a systematic analysis of gene networks in neuron types selectively vulnerable to Alzheimer's disease has suggested processes related to axon plasticity and synaptic vesicle transmission, particularly with relation to microtubule dynamics, may be involved in the neuronal vulnerability (*Roussarie et al., 2020*). Combining gene profiling of specific cell types in hippocampus with advanced technology in function dissection will continue to provide clarification to roles of DLK in the central nervous system under normal and pathological conditions.

## Materials and methods

### Key resources table

| Reagent type (species) or resource | Designation | Source or reference | Identifiers | Additional information |
|---|---|---|---|---|
| Genetic reagent (*Mus musculus*) | Conditional DLK knockout: Map3k12fl/fl | PMID:33475086; 27511108; 35361703 | | housed in UCSD vivarium |
| Genetic reagent (*M. musculus*) | Inducible DLK overexpression: H11-DLK(iOE) | PMID:33475086 | | housed in UCSD vivarium |
| Genetic reagent (*M. musculus*) | Slc17a7Cre | The Jackson Laboratory | Strain #023527; RRID:IMSR_JAX:023527 | B6;129S-Slc17a7tm1.1(cre)Hze/J |

*Continued on next page*

*Continued*

| Reagent type (species) or resource | Designation | Source or reference | Identifiers | Additional information |
|---|---|---|---|---|
| Genetic reagent (*M. musculus*) | Rpl22HA | The Jackson Laboratory | Strain #029977; RRID:IMSR_JAX:029977 | B6J.129(Cg)-Rpl22tm1.1Psam/SjJ |
| Genetic reagent (*M. musculus*) | Rosa26tdT | The Jackson Laboratory | Strain #007914; RRID:IMSR_JAX:007914 | B6.Cg-Gt(ROSA)26Sortm 14(CAG-tdTomato)Hze/J |
| Antibody | Rabbit polyclonal anti-Map3k12 antibody | Genetex | GTX124127; RRID:AB_11170703 | IF (1:250) tissue, (1:1000) cells, WB (1:1000); Lot #40653 |
| Antibody | Rabbit monoclonal anti-p-c-Jun (Ser73) (D47G9) antibody | Cell signaling | 3270; RRID:AB_2129575 | IF (1:200) tissue, Lot #5 |
| Antibody | Rabbit polyclonal anti-GFAP antibody | Dako | Z0334; RRID:AB_10013382 | IF: (1:500); Lot #20049469 |
| Antibody | Rabbit polyclonal anti-IBA1 | Wako | 019–19741; RRID:AB_839504 | IF: (1:1000) |
| Antibody | Rat monoclonal anti-HA High Affinity | Roche | 11867423001; RRID:AB_390918 | IP (5 ug); Lot #47877600 |
| Antibody | Rabbit monoclonal anti-HA (C29F4) | Cell Signaling | 3724; RRID:AB_1549585 | WB (1:1000); Lot #8 |
| Antibody | Mouse monoclonal anti-NeuN | Millipore | MAB377; RRID:AB_2298772 | IF (1:200); Lot #3104227/3808682 |
| Antibody | Mouse monoclonal anti-Tubb3 (Tuj1) | Biolegend | 801202; RRID:AB_2313773 | IF (1:1000) tissue, (1:5000) cells; Lot #B249869 |
| Antibody | Rabbit polyclonal anti-Tubb3 | Sigma-Aldrich | T2200; RRID:AB_262133 | IF (1:500) cells; Lot #21190649 |
| Antibody | Mouse monoclonal anti-Acetyl-Tubulin (6-11b-1) | Sigma-Aldrich | T7451; RRID:AB_609894 | IF (1:500) tissue, (1:3000) cells; WB (1:1000) |
| Antibody | Mouse monoclonal anti-Stmn4 | Santa Cruz Biotechnology | Sc-376936 | IF (1:250) cells; WB (1:50); Lot # E3012 |
| Antibody | Rabbit polyclonal anti-Stmn4 | Proteintech | 12027–1-AP; RRID:AB_2197401 | IF (1:400) cells; WB (1:1000); Lot#00005750 |
| Antibody | Mouse monoclonal anti-Stmn2 | R&D Systems | MAB6930; RRID:AB_10972937 | IF (1:1000) cells; WB (0.4 ng/mL); Lot#CFIL052310A |
| Antibody | Rabbit polyclonal anti-Stmn2 | Proteintech | 10586–1-AP; RRID:AB_2197283 | IF (1:400) cells; WB (1:2000); Lot#00124321 |
| Antibody | Mouse monoclonal anti-Tyrosinated Tubulin (TUB1A2) | Sigma-Aldrich | T9028; RRID:AB_261811 | IF (1:1000) tissue, (1:5000) cells; WB (1:1000); Lot #22181017 |
| Antibody | Rabbit polyclonal anti-Vglut1 | Synaptic Systems | 135 302; RRID:AB_887877 | IF (1:1000) tissue; Lot #1–53 |
| Antibody | Rabbit monoclonal anti-c-Jun (60 A8) | Cell Signaling | 9165; RRID:AB_2130165 | IF (1:200) tissue, (1:1000) cells; Lot #11 |
| Antibody | Mouse monoclonal anti-Bassoon (SAP7F407) | Novus | NB120-13249; RRID:AB_788125 | IF (1:500) tissue, cells; Lot #06082117 |
| Antibody | Chicken polyclonal anti-MAP2 | Abcam | Ab5392; RRID:AB_2138153 | IF (1:5,000) cells; Lot #1012833–1 |
| Antibody | Monoclonal mouse anti-beta actin | ABclonal | AC004; RRID:AB_2737399 | WB (1:5000); Lot #3500100012 |
| Antibody | Mouse monoclonal anti-Flag M2 | Sigma-Aldrich | F1804; RRID:AB_262044 | WB (1:500) |
| Antibody | Rabbit polyclonal anti-Homer1 | Synaptic systems | 160–003; RRID:AB_887730 | IF (1:500) tissue |
| Antibody | Mouse monoclonal anti-Satb2 | Abcam | Ab51502; RRID:AB_882455 | IF (1:500) cells |
| Antibody | Goat polyclonal anti-Prox1 | R&D Systems | AF2727; RRID:AB_2170716 | IF (4 µg/mL) |

*Continued on next page*

*Continued*

| Reagent type (species) or resource | Designation | Source or reference | Identifiers | Additional information |
|---|---|---|---|---|
| Antibody | Rabbit polyclonal anti-Map3k13 | Sigma-Aldrich | HPA016497; RRID:AB_10670027 | IF (1:200) tissue |
| Antibody | Alexafluor488 goat anti mouse IGG (H+L) | Invitrogen | A11001; RRID:AB_2534069 | IF (1:500) tissue, (1:2000) cells; Lot #745480 |
| Antibody | Alexafluor488 donkey anti mouse IGG (H+L) | Invitrogen | A21202; RRID:AB_141607 | IF (1:500) tissue, (1:2000) cells; Lot #2266877 |
| Antibody | Alexafluor647 goat anti rabbit IGG (H+L) | Invitrogen | A21245; RRID:AB_2535813 | IF (1:500) tissue, (1:2000) cells; Lot #2299231 |
| Antibody | Alexafluor488 donkey anti rabbit IGG (H+L) | Invitrogen | A21206; RRID:AB_2535792 | IF (1:500) tissue, (1:2000) cells; Lot #2376850 |
| Antibody | Alexafluor647 goat anti mouse IGG (H+L) | Invitrogen | A21236; RRID:AB_2535805 | IF (1:500) tissue, (1:2000) cells; Lot #2300995 |
| Antibody | Alexa Fluor 647 goat anti chicken IgG (H+L) | Invitrogen | A21449; RRID:AB_2535866 | IF (1:2000) cells; Lot #2079903 |
| Antibody | Anti-rabbit: ECL Anti-Rabbit IgG, HRP | Cytiva | NA934V; RRID:AB_772206 | WB (1:5000); Lot #17624274 |
| Antibody | Anti-mouse: ECL Anti-mouse IgG, HRP | Cytiva | NXA931V; RRID:AB_772209 | WB (1:5000); Lot #17675041 |
| Antibody | Stabilized goat anti-rabbit HRP conjugated | Pierce | 1858415 | WB (1:5000); Lot # HE104909 |
| Sequence-based reagent | RNAscope probe MAP3K12-C2 | ACD | ACD:458151 C2 | |
| Sequence-based reagent | RNAscope probe Slc17a7-C3 | ACD | ACD:416631 C3 | |
| Sequence-based reagent | RNAscope probe Gfap-C2 | ACD | ACD:313211 C2 | |
| Sequence-based reagent | RNAscope probe Stmn4 | ACD | ACD:537541 | |
| Commercial assay or kit | DeadEnd Fluorometric TUNEL System | Promega | G3250 | |
| Commercial assay or kit | RNAeasy Minikit | Qiagen | 74104 | |
| Commercial assay or kit | Superscript III First Strand Synthesis System | Invitrogen | 18080051 | |
| Commercial assay or kit | iQ Sybr Green Supermix | Bio-Rad | 1708880 | |
| Commercial assay or kit | Pierce BCA Protein Assay Kits | Thermo Scientific | 23227 | |
| Commercial assay or kit | RNAscope Fluorescent Multiplex Reagent kit | ACD | 320850 | Amp 4 Alt A-FL |
| Chemical compound, drug | Cycloheximide | Sigma-Aldrich | C4859 | |
| Software, algorithm | Galaxy | PMID:29790989 | RRID:SCR_006281 | https://usegalaxy.org/ |
| Software, algorithm | FastQC | Babraham Bioinformatics | RRID:SCR_014583 | https://github.com/s-andrews/FastQC |
| Software, algorithm | STAR aligner | PMID:23104886 | RRID:SCR_004463 | https://github.com/alexdobin/STAR |
| Software, algorithm | FeatureCounts | PMID:24227677 | RRID:SCR_012919 | https://subread.sourceforge.net/ |
| Software, algorithm | RStudio | Posit | RRID:SCR_000432 | https://posit.co/download/rstudio-desktop/ |
| Software, algorithm | ggplot2 | *Wickham, 2016* | RRID:SCR_014601 | https://ggplot2.tidyverse.org/ |
| Software, algorithm | DAVID | PMID:19131956 | RRID:SCR_001881 | https://david.ncifcrf.gov/home.jsp |
| Software, algorithm | Rank Rank Hypergeometric Overlap | PMID:20660011 | RRID:SCR_014024 | https://systems.crump.ucla.edu/rankrank/rankranksimple.php |
| Software, algorithm | SynGO | PMID:31171447 | RRID:SCR_017330 | https://www.syngoportal.org/ |
| Software, algorithm | GSEA | PMID:16199517 | RRID:SCR_003199 | https://www.gsea-msigdb.org/gsea/index.jsp |
| Software, algorithm | Fiji | PMID:22743772 | RRID:SCR_002285 | https://imagej.net/software/fiji/ |
| Software, algorithm | GraphPad Prism | GraphPad Software | RRID:SCR_002798 | http://www.graphpad.com |
| Other | Protein G Dynabeads | Invitrogen | 10003D | |

*Continued on next page*

*Continued*

| Reagent type (species) or resource | Designation | Source or reference | Identifiers | Additional information |
|---|---|---|---|---|
| Other | DAPI | Invitrogen | D1306 | |
| Other | B27 | Gibco | 17504–044 | |
| Other | RNAse inhibitor, murine | New England Biolabs | M0314 | |

## Experimental mice

All animal protocols were approved by the Animal Care and Use Committee of the University of California San Diego. Map3k12fl (*Map3k12*<sup>fl/fl</sup>) allele was made by Dr. Lawrence B. Holzman (Univ. Penn) and reported in *Chen et al., 2016*; *Li et al., 2021*; *Saikia et al., 2022*. Map3k12 (H11-DLK(iOE)) transgene was described in *Li et al., 2021*. Slc17a7<sup>Cre</sup> allele (JAX stock #023527) was described in *Harris et al., 2014*. RiboTag allele (JAX stock #029977) was described in *Sanz et al., 2009*. ROSA26-loxP-STOP-loxP-tdTomato fl/fl reporter line (JAX stock #007914) was constructed in *Madisen et al., 2010*. Standard mating procedure was followed to generate *Slc17a7*<sup>Cre/+</sup>;*Map3k12*<sup>fl/fl</sup> and *Slc17a7*<sup>Cre/+</sup>;H11-DLK(iOE)/+ experimental mice. Genotyping primers are in *Supplementary file 4*. Sibling control mice had either *Slc17a7*<sup>Cre/+</sup> or *Map3k12*<sup>fl/fl</sup> or H11-DLK(iOE)/+ allele alone. All experiments used both male and female mice. *Slc17a7*<sup>Cre</sup> dependent tdTomato expression from H11-DLK(iOE) transgene was observed in most or all CA3, many CA1 neurons, with limited number of DG neurons at P15, similar to the described *Slc17a7*<sup>Cre</sup> reporter line (*Harris et al., 2014*), and was throughout all regions by P60. *Slc17a7*<sup>Cre/+</sup>;H11-DLK(iOE)/+ mice around 4 months of age developed noticeable progressive motor deficits, which were likely unrelated to hippocampal glutamatergic neuron death, and were not studied further.

## Western blotting

Isolated hippocampal tissue was lysed in ice-cold RIPA buffer (50 mM Tris/HCl pH 7.4, 150 mM NaCl, 0.5% DOC, 0.1% SDS, 1% NP-40 freshly supplemented with protease inhibitor cocktail and 1 mM PMSF). Tissues were homogenized by Dounce homogenization using 30 passes pestle A and 30 passes pestle B. Samples were spun down at 13,000 x *g* for 10 min at 4 C. Supernatants were collected, and protein concentration was determined using the BCA assay (Thermo Fisher Scientific, 23227). Equal concentration of proteins (~10–20 ng) were run on NuPAGE 4–12% Bis-Tris Gel, 1.0 mm (Invitrogen, NP0322BOX) with 20X NuPAGE MES SDS Running Buffer (Invitrogen, NP0002). Protein samples were transferred by wet transfer to a PVDF membrane (0.2 µm, Bio-RAD, 1620177) by Mini Trans-Blot Cell at 100 mA for 1 hr at 4 °C. Membranes were blocked in 5% skim milk in TBST for 1 hr at room temperature, and then incubated with primary antibody in 3% BSA or 5% skim milk in TBST at 4 °C overnight. Membranes were washed 3x10 min in TBST and incubated with 1:5000 of the appropriate HRP-conjugated secondary antibody in 3% BSA in TBST at room temperature for 1 hr, then washed 3x10 min in TBST. Bands were detected using enhanced chemiluminescence (ECL) reagents (GE Healthcare, RPN2106) or Pico PLUS Chemiluminescent Substrate (Thermo Fisher Scientific, 34580) using a Licor Odyssey XF Imager. Molecular weight markers were PageRuler Plus Prestained Protein Ladder (Thermo Fisher Scientific, 26619) or Precision Plus Protein Ladder (Bio-Rad, 1610374).

Quantification of western blot images was performed by measuring identical size regions from each band, subtracting the background signal, and normalizing to internal actin controls for each sample. Time course analysis was further normalized to P1 WT protein levels. All images shown had N=3 biological replicates.

## STMN2/STMN4 antibody specificity

Given their highly similar protein size and sequences, we wanted to evaluate STMN2 and STMN4 antibody specificity. We used two antibodies for each. STMN2 antibodies were a mouse monoclonal anti-STMN2 (R&D Systems, MAB6930) and a rabbit polyclonal anti-STMN2 (Proteintech, 10586–1-AP). STMN4 antibodies were a mouse monoclonal anti-STMN4 (Santa Cruz, Sc-376936) and a rabbit polyclonal anti-STMN4 (Proteintech, 12027–1-AP). We tested the specificity of STMN2 and STMN4 antibodies by co-staining for STMN2 and STMN4, or with two separate STMN2 or STMN4 antibodies. In each case, antibody signal overlapped in the cell soma, presumably at the Golgi, as well as larger puncta elsewhere, with some overlapping small puncta, and some non-overlapping small puncta

(*Figure 6—figure supplement 1A*). Overlapping and non-overlapping signal was also visualized by plotting intensity of signal along the neurite. By western blot STMN2 and STMN4 were highly similar at the protein MW and levels, though the STMN4 antibodies tested detected a larger MW band specific to STMN4, suggesting some specificity. The STMN2 antibodies also occasionally recognized a smaller MW band only recognized with the Proteintech STMN4 antibody but not the Santa Cruz STMN4 antibody (*Figure 6—figure supplement 1B*). Furthermore, while STMN4 protein levels increased relative to β-actin in mice with increased DLK, STMN2 protein levels did not show significant increases. These different expression patterns validated some degree of specificity with these antibodies. Based on our analysis, the STMN4 Santa Cruz antibody (Sc-376936) may be more specific to STMN4 than the STMN4 Proteintech antibody (12027–1-AP) though it appears less sensitive. The STMN2 antibodies showed strongest overlap of puncta and similar MW proteins, thus we were unable to detect differences in specificity. Whether the antibodies may also detect some of the same isoforms is not clear without further analysis.

## Immunofluorescence of hippocampal tissues

Mice were transcardially perfused with saline solution followed by 4% PFA in PBS. Brains were dissected and post-fixed overnight in 4% PFA at 4 °C, then washed with PBS and transferred to 30% sucrose in PBS for at least three days. Brains were mounted coronally for cryosectioning in OCT Compound (Fisher HealthCare, 4585) on dry ice. Sections were cut to 25 µm thickness, divided evenly among six wells, and stored in PBS with 0.01% sodium azide at 4 °C until staining. For immunostaining, free floating sections were washed 3 times in 0.2% Triton X-100 in PBS, blocked for 1 hr at room temperature in 5% donkey serum in 0.4% Triton X-100 in PBS, then incubated with primary antibodies in 2% donkey serum in 0.4% Triton X-100 in PBS overnight at 4 °C rocking. Following three washes with 0.2% Triton in PBS, sections were incubated with secondary antibodies in 2% donkey serum in 0.4% Triton X-100 in PBS for 1 hr at room temperature. Sections were again washed three times with 0.2% Triton X-100 in PBS, stained with DAPI for 10 min (14.3 mM in PBS) and washed three times in PBS before mounting on glass slides using Prolong Diamond Antifade Mountant. TUNEL staining was performed using the DeadEnd Fluorometric TUNEL System (Promega, G3250) with a modified protocol as described previously (*Li et al., 2021*).

## Immunoprecipitation and isolation of ribosome associated mRNA

Immunoprecipitation of HA-tagged ribosomes was conducted following the protocol described in *Sanz et al., 2019*. Briefly, hippocampi from both hemispheres were dissected in ice cold PBS from mice of desired genotypes at postnatal day 15, and were stored at –80 °C before further processing. Frozen tissues were homogenized by Dounce homogenization using 30 passes pestle A and 30 passes pestle B in 1.5 mL homogenization buffer (50 mM Tris, pH 7.5, 1% NP-40, 100 mM KCl, 12 mM MgCl$_2$, 100 µg/mL cycloheximide, cOmplete EDTA-free protease inhibitor cocktail (Roche), 1 mg/mL heparin, 200 U/mL RNasin, 1 mM DTT). Following centrifugation at 10,000 x *g* for 10 min at 4 °C, 5 µg anti-HA high affinity (Roche) were added to the supernatant and incubated 4 hours rotating end-over-end at 4 °C. The entire antibody-lysate solution was added to 400 µl Protein G Dynabeads per sample overnight rotating end-over-end at 4 °C. High salt buffer was prepared (50 mM Tris, pH 7.5, 1% NP-40, 300 mM KCl, 12 mM MgCl$_2$, 100 µg/mL cycloheximide, 0.5 mM DTT), and beads were washed 3x10 min using a magnetic tube rack. During the final wash, samples were transferred to a new tube, and beads were eluted in 350 µl of RLT buffer (from the Qiagen RNAeasy Minikit) supplemented with 1% β-mercaptoethanol. RNA was extracted following manufacturer's instructions in the RNAeasy Minikit (QIAGEN). RNA integrity was measured using an Agilent TapeStation conducted at the IGM Genomics Center, University of California, San Diego, La Jolla, CA. All RNA for sequencing had RIN ≥8.0, 28 S/18S≥1.0.

To confirm immunoprecipitation in RiboTag IP samples, 10% of IP sample was isolated after final wash in high salt buffer. After removal of high salt buffer, proteins were eluted in 2X RIPA buffer and 4X Laemmli Sample Buffer (Bio-Rad, 161–0747) by heating 10 min at 50 °C. Beads were separated using a magnetic tube rack, the supernatant was isolated and beta-mercaptoethanol was added. Samples were boiled at 95 °C for 10 min and centrifuged 5 min at 13,000 x *g*. Immunoprecipitated samples were separated by SDS-PAGE using Any kD Mini-PROTEAN TGX Precast Protein Gels (Bio-Rad, 4569034).

To ensure appropriate depletion of transcripts from non-Slc17a7 expressing cells, we performed qRT-PCR analysis on representative marker genes for cell types in immunoprecipitated glutamatergic neuron RNA relative to whole hippocampal RNA. Briefly, RNAs isolated from whole hippocampi and immuoprecipitated from glutamatergic neurons were reverse transcribed to cDNA using Superscript III First Strand Synthesis System (Invitrogen, cat#18080051) following the manufacturer's protocol. 100 ng RNA/sample was reverse transcribed with random hexamers. iQ Sybr Green Supermix (Bio-Rad, #1708880) was used for qPCR, and mRNA levels of marker genes (*Slc17a7* (glutamatergic neurons), *Wfs1* (CA1 neurons), *Gfap* (Astrocytes), and *Vgat* (inhibitory neurons)) were normalized to *Gapdh* expression. Expression levels of qRT-PCR samples were analyzed using the CFX Real-Time PCR Detection System and CFX Manager Software (Bio-Rad). Relative enrichment of marker genes was evaluated using the comparative $C_T$ method. All samples were run in triplicate. Primers for *Gapdh*, *Slc17a7*, *Wfs1*, *Gfap*, and *Vgat* are from *Furlanis et al., 2019*; *Supplementary file 4*.

## Sequencing

Library preparation and sequencing for ribosome associated mRNAs were performed by the UCSD IGM Genomics Center using Illumina Stranded mRNA Prep. Sequencing was performed on NovaSeq S4 with PE100 reads.

## Read mapping

Following paired end RNA sequencing of isolated RNA, >24 million reads per sample were obtained (n=3DLK(iOE)/3 WT, n=4 DLK(cKO)/4 WT). The Galaxy platform was used for read mapping and differential expression analysis (*Afgan et al., 2018*). Read quality was checked using FastQC (version 0.11.8). Reads were mapped to the mouse reference genome (mm10) using STAR galaxy version 2.6.0b-1 with default settings (*Dobin et al., 2013*). Four DLK(cKO) and controls included 2 male and 2 female. For DLK (iOE), one female sample was removed from each genotype control and DLK (iOE) due to read mapping variability/read quality, resulting in N=3 per genotype (2 male/1 female). Mapped reads were assigned to genes using featureCounts version 1.6.3 (*Liao et al., 2014*). High Pearson correlation (*r*>0.99) was observed between all *Slc17a7$^{Cre/+}$;Map3k12$^{fl/fl}$;Rpl22$^{HA/+}$* or *Slc17a7$^{Cre/+}$;*H11-DLK(iOE)/+;Rpl22$^{HA/+}$* samples and their respective littermate controls. Differential gene expression analysis was conducted using DESeq2 galaxy version 2.11.40.2 (*Love et al., 2014*) with genotype, sex, and batch included as factors in the analysis. Generation of volcano plots was performed in RStudio version 1.2.1335 using the ggplot2 package version 3.3.5 (*Wickham, 2016*). Heatmaps were generated using the heatmap.2 function on Galaxy (Galaxy version 3.0.1) using normalized gene counts with a log2 transformation and scaling by row.

## Gene ontology

Gene ontology analysis was performed using DAVID 2021 version (*Huang et al., 2009*; *Sherman et al., 2022*) on genes found to be differentially expressed with <0.05. For gene ontology and pathway analysis, background gene lists were generated by removing any gene with a base mean from DEseq2 normalization less than 1. *Gfap* was removed from GO and pathway analysis as a differentially expressed gene as it likely reflected a small amount of contamination from non-Slc17a7-positive cells. DAVID analysis was performed using default thresholds, and Benjamini corrected p-values are reported. GO terms displayed in figures were chosen from top terms reaching significance related to biological processes or cellular components (BP5, CC4 or CC5) categories after filtering terms for semantic similarity. For SynGO analysis, mouse genes detected as differentially expressed were converted to human IDs using the ID conversion tool, and analysis was performed using the brain expressed background gene list provided by SynGO (*Koopmans et al., 2019*; Version/release 20210225).

## Rank rank hypergeometric overlap (RRHO) analysis for correlation of gene expression patterns

We used Rank Rank Hypergeometric overlap (https://systems.crump.ucla.edu/rankrank/rankrank-simple.php) to compare DLK(iOE) and DLK(cKO) translatome datasets (*Plaisier et al., 2010*). Input gene lists included 12740 genes which were expressed across all samples. For each gene, the -Log10P$_{adj}$ was multiplied by the sign of the fold change to obtain the metric used for ranking. Both

DLK(iOE) and DLK(cKO) datasets were ranked in order to have increasing DLK along the x and y axis. RRHO was run using a step size of 100 genes. The Benjamini-Yekutieli corrected graph is shown.

## Hippocampal spatial expression analysis

Comparison with gene expression databases: False color expression images from the Allen Mouse Brain Atlas were used for evaluating expression pattern, and numbers were assigned based on color in dorsal hippocampus (Red = 3, Yellow = 2, Blue/Green = 1, No = 0). When intensity varied across sections or intensity was in-between two categories, preference was given to depicting general patterns of relative expression over absolute signal. When in situ data was not available, or expression patterns were unclear, we used additional transcriptomic data to assess spatial expression (*Habib et al., 2016*; *Zeisel et al., 2018*), and values were chosen to reflect relative expression. Generally, the following scale was used for *Habib et al., 2016* data through the Single Cell Portal: 0 if next to no signal, 1 if expression in some cells, but average was still zero, 2 if quartile 3 value in violin plot is >0, 3 if higher average signal, again values were chosen to reflect relative expression. Genes were categorized as enriched in a region/s if one or two regions show higher values than another region. If two regions show different expression levels but are two levels above third region, the gene is considered as enriched in both (i.e. CA1=2, CA3=3, DG = 0, considered as CA1, CA3 enriched). If only one level above other regions, the gene is enriched only in the region with strongest expression (i.e. CA1=3, CA3=2, DG = 1, considered as CA1 enriched). Most expressed elsewhere in hippocampus used when the strongest expression is found in another region/cell type, and other descriptions do not explain where most of the signal is.

Comparison with CamK2-RiboTag and Grik4-RiboTag data: Gene set enrichment analysis (GSEA) (4.2.2) (*Subramanian et al., 2005*) was performed on Slc17a7-RiboTag expression data after filtering lowly expressed genes using normalized counts. Analysis was conducted using the parameters: 1000 permutations, no collapse gene set, and permutation type gene set, with all other settings as default. To define gene sets for CA1 or CA3 enriched genes, we analyzed RiboTag datasets (*Traunmüller et al., 2023*; GSE209870) in wild type 6-week-old CA1 and CA3 neurons, from CamK2-cre and Grik4-cre mice, respectively. We compared the CamK2-RiboTag dataset and Grik4-RiboTag dataset to identify genes which were enriched in CA1 compared to CA3 or vice versa, applying an expression filter (average of at least 50 reads/animal) to ensure genes enriched in a particular region were expressed. The top 100 genes enriched in CamK2-RiboTag relative to Grik4-RiboTag were considered 'CA1 genes'. The top 100 genes enriched in Grik4-RiboTag relative to CamK2-RiboTag were considered 'CA3 genes'. 82 out of 100 GRIK4 (CA3) and 83 out of 100 CAMK2 (CA1) enriched genes were expressed in both our WT and DLK(cKO) samples (*Supplementary file 3* CamK2 Grik4 enriched genes).

## RNAscope analysis

The RNAscope Fluorescent Multiplex Reagent kit (Amp 4 Alt A-FL, Cat. #320850) (*Wang et al., 2012*) with probes from Advanced Cell Diagnostics were used. The protocol was carried out under RNase-free conditions and following the manufacturer's instructions. Mice were anesthetized with isoflurane prior to decapitation. Brains were dissected immediately and flash frozen in OCT at –80 °C. Fresh-frozen tissue was cryosectioned coronally to 20 µm, collected on glass slides (Superfrost Plus), and stored at –80 °C. Slides were fixed with 4% paraformaldehyde, dehydrated with 50% ethanol, 70% ethanol, and 2 x washes in 100% ethanol for 5 min each at RT, followed by incubation in Protease IV reagent for 30 min at 40 °C. Hybridization with target probes was performed at 40 °C for 2 hr in a humidified slide box in an incubator followed by wash and amplification steps according to the manufacturer's protocol. Finally, tissue was counterstained with DAPI, and mounted with Prolong diamond antifade mountant. All target probes were multiplexed with probes for *Slc17a7* to label glutamatergic neurons.

## Primary hippocampal neuron cultures and immunostaining

Prior to preparing cultures, Poly-D-Lysine (Corning, Cat#354210) was coated on 12 mm glass coverslips (0.2 mg/mL) or six-well plates (0.05 mg/ml) for 2 days at 37 °C. Neurons with indicated genotypes were labeled by tdTomato from Cre-dependent Rosa26-tdTomato generated from the following crosses: for control: *Slc17a7^{Cre/+}* X *Rosa26^{tdT/+}*; for DLK(cKO): *Slc17a7^{Cre/+};Map3k12^{fl/}*

$^{fl}$ X *Map3k12$^{fl/fl}$;Rosa26$^{tdT/+}$*; for DLK(iOE): H11-DLK(iOE)/H11-DLK(iOE) X *Slc17a7 $^{Cre/+}$;Rosa26$^{tdT/+}$*. Primary neurons were generated from hippocampi of P1 pups. Mice were rapidly decapitated, then brains were removed, placed into ice cold HBSS (calcium- and magnesium-free) supplemented with 10 mM HEPES for removal of meninges and dissection of hippocampi (*Kaech and Banker, 2006*). Dissected hippocampi were dissociated in HBSS with HEPES in 0.25% trypsin for 15 min at 37 °C, and were then washed 3 times with 5 ml of 20% Fetal bovine serum in HBSS. Dissociated cells from a litter were pooled into the same culture. Cells were triturated in Opti-MEM supplemented with 20 mM glucose by five passes with an unpolished glass pipette and five to ten passes using a fire polished glass pipette. Cells were counted using a hemocytometer, and 60,000 cells were plated per coverslip into a 24-well plate or 300,000 per well of a six-well dish. Cultures were kept in an incubator at 37 °C with 5% CO2. After 4 hr, plating media was replaced with prewarmed Neurobasal Medium supplemented with glutamine, penicillin/streptomycin, and B27. Cells were fixed after 48 hr (DIV2) or on DIV14 with prewarmed 4% PFA/4% sucrose in PBS for 20 min at room temperature followed by three washes with PBS. Media were changed carefully to minimize impacts to growth cone morphology.

Staining of fixed neurons was performed in 24-well plates. Coverslips were incubated in 50 mM ammonium chloride for 10 min, followed by three washes PBS, 5 min 0.1% Triton X-100 in PBS, and blocking in 30 mg/ml Bovine serum albumin (BSA) in 0.1% Triton in PBS for 30 min. Coverslips were incubated in primary antibody diluted in 30 mg/ml BSA in 0.1% Triton in PBS according to antibody table for 90 min at room temperature followed by four washes in 0.1% Triton in PBS. Secondary antibodies were diluted in 30 mg/ml BSA in 0.1% Triton in PBS with 1% donkey serum according to antibody table, and incubated for 60 min at room temperature. Finally, coverslips were washed three times in 0.2% Triton in PBS, stained with DAPI, washed three times with PBS, and mounted using Prolong Diamond Antifade Mountant. For an unknown reason, co-immunostaining of DLK and tyrosinated tubulin led to a pattern of DLK staining different from the punctate appearance of DLK observed in other conditions. The typical appearance of DLK could still be observed in cells with high levels of DLK. This altered appearance was not observed during co-immunostaining of DLK and acetylated tubulin.

## Confocal imaging and quantification

Fluorescent images were acquired using a Zeiss LSM800 confocal microscope using a 10x, 20x, or 63x objective. All tissue sections and neurons within the same experiment were imaged under identical conditions. For brain tissue, three sections per mouse were imaged with a minimum of three mice per genotype for data analysis. Dorsal hippocampal images were taken from approximately bregma –1.5 mm to –2.3 mm. For image analysis, the quantification was performed blind to genotype or in an automated manner when possible. All image processing and analysis was performed using Fiji distribution of ImageJ unless otherwise specified (*Schindelin et al., 2012*).

For quantification of mRNA puncta, ROI were drawn to count puncta overlapping with nuclei of *Slc17a7*-positive cells. Individual puncta were counted from >50 cells per genotype in a blinded manner. Puncta counts were normalized to *Slc17a7* puncta counts to control for variability in staining or preservation of RNA. Three to four sections per mouse were quantified and three mice per genotype were stained with each probe.

Pyramidal cell layer thickness was measured across CA1 by averaging the lengths of three perpendicular lines extending across the maximum projected z-stack of the pyramidal cell layer for each section. Three sections were averaged per mouse from dorsal hippocampus. For sections including ventral hippocampus, cell layer thickness of CA1 was measured using three lines either above the ventral edge of the suprapyramidal blade of dentate gyrus (*Dong et al., 2009*) for dorsal hippocampus (posterior) quantifications or below the ventral edge of the DG for ventral CA1 quantifications. Hippocampal cross-sectional area was measured by tracing outlines of CA1, CA3, or DG (including dendritic layers) in dorsal hippocampus sections.

DLK signal intensity in immunofluorescence images was quantified by drawing an outline around CA1, CA3, or DG (all cell layers), and measuring the mean fluorescence intensity.

Tuj1, tyrosinated tubulin, acetylated tubulin, and MAP2 intensities were measured using the mean gray value from auto thresholding (default) over stratum radiatum of CA1, the molecular layer of DG, or stratum lacunosum-moleculare, stratum radiatum, and stratum lucidum of CA3.

Staining of p-c-Jun in P60 animals and c-Jun from all timepoints was quantified from 20x images using mean gray values of ROIs for each brain slice cropped around the pyramidal cell or granule cell layers with background subtraction of non-nuclear signal from dendritic regions. Analysis of p-c-Jun-positive nuclei in DLK(iOE) mice was counted from 10x images with using an intensity threshold of 20000 (P10) or 110 or 140 (P15) depending on imaging conditions. Nuclei were separated using a watershed, and all nuclei larger than 10 µm² were counted.

TUNEL positive signals were counted as fluorescent signals overlapping with the pyramidal cell or dentate granule cell layer in each region from 10x tile scan images of dorsal hippocampus. Z-stacks covering the entire section were max projected for quantification.

VGLUT1, Bassoon, and Homer1 puncta were quantified from stratum radiatum of dorsal CA1. Images were quantified using a single slice image, and a 25x25 µm ROI was chosen to minimize absence of puncta due to cell bodies. A gaussian filter of 1 pixel was applied to the image. Background subtraction was performed using a rolling ball radius of 10 pixels, and an automated threshold was applied to the image using the Otsu method followed by a watershed to separate clustered puncta. Puncta larger than 2 pixels were counted for individual proteins. Overlap of Bassoon and Homer1 puncta of any size were counted. The number and average size of puncta were recorded from two images per brain section and three sections per mouse.

GFAP mean fluorescence intensity was quantified in a 312µm x 312µm box around the pyramidal cell or granule cell layers of CA1, CA3, and DG with background intensity subtracted after measuring from an area without GFAP signal.

Neurons were selected for neurite outgrowth and axon analysis after confirming DLK protein level by antibody staining and measurement of DLK fluorescence intensity in cell soma at DIV2. While we used tdTomato as a reporter for Slc17a7-positive neurons, not all tdTomato-positive neurons showed detectable differences in DLK levels at this early (DIV2) timepoint. Cell somas were outlined using tdTomato, and DLK integrated density was measured. Integrated density reflected the mean gray value multiplied by the area. *Slc17a7*<sup>Cre/+</sup> control cells were selected for further analysis if DLK integrated density was 4000–8500. Cells from DLK(cKO) cultures with integrated density values of DLK less than 4000 were selected for further analysis as 'DLK(cKO)' and cells from DLK(iOE) with integrated density values of greater than 8500 were selected for further analysis as 'DLK(iOE)'. While we observed variably increased DLK signals in DLK(iOE) neurons from moderate to strong, all DLK(iOE) neurons with increased levels above the set threshold were grouped together in quantifications due to limited numbers of neurons. Primary neurites were counted in a blinded manner from tdTomato channel, counting both branches and filopodia originating from cell soma region. Neurites were considered as axons in axon specification analysis if longer than 90 µm.

Bassoon puncta in cell culture were quantified from 20 µm stretches of neurites. Regions for analysis were selected based on tdTomato-positive signal on thin processes without dendritic spines exhibiting Bassoon signal, that was not in a region densely populated by Bassoon signal from other neurites. DLK levels were also used to select ROIs. Signal from tdTomato was used to create a 10 pixel ROI along the neurite. Bassoon puncta were identified in a blinded manner by smoothing the image, applying a triangle threshold, and manually dividing merged puncta based on bassoon intensity. All puncta 5 pixels or larger and overlapping with tdTomato signal were analyzed for puncta size and density.

Dendritic spines were quantified from a 20 µm countable and representative stretch of dendrite within 75 µm of the neuron soma from one of the three largest dendrites. Spine density was counted using tdTomato signal, and calculated by counting total dendritic spines divided by the traced length of dendrite. Spines were manually categorized following measurements in *Risher et al., 2014*. Filopodia (>2 µm) are not included in spine density counts. Spines were quantified from independent cultures per genotype with 8–16 neurons per culture.

## Stastical analysis

All statistical analysis shown in graphs was performed using GraphPad Prism 9.4.0. Points represent individual values, with bars reflecting mean values, and error bars plotting standard error of the mean (SEM).

## Acknowledgements

We thank members of our labs for their support and valuable discussion throughout this work. We are grateful to Emily Griffin in Susan Ackerman's lab and Caitlin Rodriguez in Aaron Gitler's lab for advice on troubleshooting immunoprecipitation of ribosomes for RiboTag, to Brenda Bloodgood for advice with RNAscope experiments, Gentry Patrick, Lara Dozier, and Frank Bradke for their guidance in primary hippocampal cultures, Megan Williams and Stacey Glasgow for advice and CA1 and CA3 neuron antibodies, and Gareth Thomas for discussion and comments. This publication includes data generated at the UC San Diego IGM Genomics Center utilizing an Illumina NovaSeq 6000 that was purchased with funding from a National Institutes of Health SIG grant (#S10 OD026929). DA was supported by the TÜBİTAK 2214 A International Research Fellowship Programme. EMR received an Innovative Research Grant from the Kavli Institute for Brain and Mind. This work was supported by a grant from NIH (NS R35 127314 to YJ).

## Additional information

### Funding

| Funder | Grant reference number | Author |
| --- | --- | --- |
| National Institute of Neurological Disorders and Stroke | R35 127314 | Yishi Jin |
| Kavli Institute for Brain and Mind, University of California, San Diego | 2020-1711 | Erin M Ritchie |
| TÜBİTAK | 2214-A | Dilan Acar |

The funders had no role in study design, data collection and interpretation, or the decision to submit the work for publication.

### Author contributions

Erin M Ritchie, Conceptualization, Data curation, Formal analysis, Investigation, Methodology, Writing – original draft, Writing – review and editing; Dilan Acar, Siming Zhong, Qianyi Pu, Formal analysis, Investigation; Yunbo Li, Methodology; Binhai Zheng, Resources, Writing – review and editing; Yishi Jin, Conceptualization, Resources, Funding acquisition, Investigation, Writing – original draft, Project administration, Writing – review and editing

### Author ORCIDs

Erin M Ritchie ⬤ https://orcid.org/0000-0002-3558-3029
Yishi Jin ⬤ https://orcid.org/0000-0002-9371-9860

### Ethics

All of the animals were handled according to approved institutional animal care and use committee (IACUC) protocols (#S13072) of the University of California San Diego.

Reviewer #1 (Public review): https://doi.org/10.7554/eLife.101173.3.sa1
Reviewer #2 (Public review): https://doi.org/10.7554/eLife.101173.3.sa2
Author response https://doi.org/10.7554/eLife.101173.3.sa3

## Additional files

### Supplementary files

Supplementary file 1. Excel file containing DLK(cKO) differentially expressed genes. File containing differential expression results from DLK(cKO) compared to control and their regional enrichment. Highlighted columns show gene symbols of differentially expressed genes ($p_{adj}$ <0.05), Log2 fold change, and adjusted p-values. Sheets show genes sorted by all differentially expressed genes, upregulated genes, downregulated genes, and synaptic genes.

Supplementary file 2. Excel file containing DLK(iOE) differentially expressed genes. File containing differential expression results from DLK(iOE) compared to control and their regional enrichment. Highlighted columns show gene symbols of differentially expressed genes ($p_{adj}$ <0.05), Log2 fold change, and adjusted p-values. Sheets show genes sorted by all differentially expressed genes, regional expression, upregulated genes, downregulated genes, and synaptic genes.

Supplementary file 3. Excel file containing CamK2 and Grik4 RiboTag enriched genes. File containing top 100 genes enriched in CamK2-RiboTag compared to Grik4-RiboTag and vice versa (*Traunmüller et al., 2023*; GSE209870) as described in methods.

Supplementary file 4. Primers used for genotyping and qRT-PCR. File containing primer sequences used for genotyping and qRT-PCR.

MDAR checklist

## Data availability

Sequencing data have been deposited in GEO under accession code GSE266662.

The following dataset was generated:

| Author(s) | Year | Dataset title | Dataset URL | Database and Identifier |
|---|---|---|---|---|
| Ritchie EM, Jin Y | 2025 | DLK-dependent protein network regulates hippocampal glutamatergic neuron degeneration | https://www.ncbi.nlm.nih.gov/geo/query/acc.cgi?acc=GSE266662 | NCBI Gene Expression Omnibus, GSE266662 |

The following previously published dataset was used:

| Author(s) | Year | Dataset title | Dataset URL | Database and Identifier |
|---|---|---|---|---|
| Traunmueller L, Scheiffele P | 2022 | Trans-cellular regulation of synaptic properties by neuron-specific alternative splicing | https://www.ncbi.nlm.nih.gov/geo/query/acc.cgi?acc=GSE209870 | NCBI Gene Expression Omnibus, GSE209870 |

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
