## [Editor Report · eLife Assessment]

This manuscript describes the impact of modulating signaling by a key regulatory enzyme, Dual Leucine Zipper Kinase (DLK), on hippocampal neurons. The results are interesting and will be **important** for scientists interested in synapse formation, axon specification, and cell death. The authors have carefully addressed the comments made by the reviewers and the findings are **convincing** in large part due to the use of extensive mouse genetics, detailed gene expression of enriched genes, and recognition of neuron vulnerability.

---

## [Referee Report · Reviewer #1 (Public review)]

Summary:

In this work Ritchie and colleagues explore functional consequences of neuronal over-expression or deletion of the MAP3K DLK that their labs and others have strongly implicated in both axon degeneration, neuronal cell death, and axon regeneration. Their recent work in eLife (Li, 2021) showed that inducible over-expression of DLK (or the related LZK) induces neuronal death in the cerebellum. Here, they extend this work to show that inducible over-expression in Vglut1+ neuron also kills excitatory neurons in hippocampal CA1, but not CA3. They complement this very interesting finding with translatomics to quantify genes whose mRNAs are differentially translated in the context of DLK over-expression or knockout, the latter manipulation having little to no effect on the phenotypes measured. The authors note that several genes and pathways are differentially regulated according to whether DLK is over-expressed or knocked out. They note DLK-dependent changes in genes related to synaptic function and to the cytoskeleton and ultimately relate this in cultured neurons to findings that DLK over-expression negatively impacts synapse number and changes microtubules and neurites, though with a less obvious correlation.

Strengths:

Where this work represents a conceptual advance is in defining DLK-dependent changes in translation. Moreover, the finding that DLK may differentially impact neuronal death will become the basis for future studies exploring whether DLK contributes to differential neuronal susceptibility to death, which is a broadly important topic.

Comments on the latest version:

The addition of the P10 data is an important advance. With this, the authors have satisfactorily addressed the concerns that I raised.

---

## [Referee Report · Reviewer #2 (Public review)]

This manuscript describes the impact of deleting or enhancing the expression of the neuronal-specific kinase DLK in glutamatergic hippocampal neurons using clever genetic strategies, which demonstrates that DLK deletion had minimal effects while overexpression resulted in neurodegeneration in vivo. To determine the molecular mechanisms underlying this effect, ribotag mice were used to determine changes in active translation which identified Jun and STMN4 as DLK-dependent genes that may contribute to this effect. Finally, experiments in cultured neurons were conducted to better understand the in vivo effects. These experiments demonstrated that DLK overexpression resulted in morphological and synaptic abnormalities.

Strengths:

This study provides interesting new insights into the role of DLK in the normal function of hippocampal neurons. Specifically, the study identifies:

(1) CA1 vs CA3 hippocampal neurons have differing sensitivity to increased DLK signaling.

(2) DLK-dependent signaling in these neurons is similar to but distinct from the downstream factors identified in other cell types, highlighted by the identification of STMN4 as a downstream signal.

(3) DLK overexpression in hippocampal neurons results in signaling that is similar to that induced by neuronal injury.

The study also provides confirmatory evidence that supports previously published work through orthogonal methods, which adds additional confidence to our understanding of DLK signaling in neurons. Taken together, this is a useful addition to our understanding of DLK function.

Comments on the latest version:

The authors have sufficiently addressed all issues raised with the initial manuscript.

---

## [Author Response]

The following is the authors’ response to the original reviews.

**eLife Assessment**
This manuscript describes the impact of modulating signaling by a key regulatory enzyme, Dual Leucine Zipper Kinase (DLK), on hippocampal neurons. The results are interesting and will be important for scientists interested in synapse formation, axon specification, and cell death. The methods and interpretation of the data are solid, but the study can be further strengthened with some additional studies and controls.

We greatly appreciate the thorough review and thoughtful suggestions from the reviewers and editors on our original manuscript. We provide point-to-point response below. We added new studies on P10 mice and controls as suggested, and made revision of figures and texts for clarification. The revised manuscript includes three new supplemental figures; major text revision is copied under response.

**Reviewer #1 (Public Review):**
Summary:In this work, Ritchie and colleagues explore functional consequences of neuronal over-expression or deletion of the MAP3K DLK that their labs and others have strongly implicated in both axon degeneration, neuronal cell death, and axon regeneration. Their recent work in eLife (Li, 2021) showed that inducible over-expression of DLK (or the related LZK) induces neuronal death in the cerebellum. Here, they extend this work to show that inducible over-expression in Vglut1+ neurons also kills excitatory neurons in hippocampal CA1, but not CA3. They complement this very interesting finding with translatomics to quantify genes whose mRNAs are differentially translated in the context of DLK over-expression or knockout, the latter manipulation having little to no effect on the phenotypes measured. The authors note that several genes and pathways are differentially regulated according to whether DLK is over-expressed or knocked out. They note DLK-dependent changes in genes related to synaptic function and the cytoskeleton and ultimately relate this in cultured neurons to findings that DLK over-expression negatively impacts synapse number and changes microtubules and neurites, though with a less obvious correlation.Strengths:This work represents a conceptual advance in defining DLK-dependent changes in translation. Moreover, the finding that DLK may differentially impact neuronal death will become the basis for future studies exploring whether DLK contributes to differential neuronal susceptibility to death, which is a broadly important topic.

We thank the reviewer for the comments on the value of our work.

Weaknesses:This seems like two works in parallel that the authors have not yet connected. First is that DLK affects the translation of an interesting set of genes, and second, that DLK(OE) kills some neurons, disrupts their synapses, and affects neurite growth in culture.Specific questions:(1) Is DLK effectively knocked out? The authors reference the floxxed allele in their 2016 work (PMID: 27511108), however, the methods of this paper say that the mouse will be characterized in a future publication. Has this ever been published? The major concern is that here the authors show that Cre-mediated deletion results in a smaller molecular weight protein and the maintenance of mRNA levels.

We apologize for out-of-date citation of the DLK(cKO)^fl/fl^ mice. The DLK(cKO)^fl/fl^ mice have been published in (Li et al., 2021; Saikia et al., 2022); excision of the flox-ed exon was verified using several Cre drivers (Pv-Cre, AAV-Cre, and VGlut1-Cre in this study). The flox-ed exon contains the initiation ATG and 148 amino acids. By western blot analysis using antibodies against C-terminal peptides of DLK on cerebellar extracts (in Li et al., 2021) and hippocampal extracts (this study), the full-length DLK protein was significantly reduced (Fig 1A-B); DLK is expressed in other hippocampal cells, in addition to glutamatergic neurons, explaining remaining full-length DLK detected.

Our Ribo-seq of VGlut1-Cre; DLK(cKO)^fl/fl^ detected remaining *Dlk* mRNAs lacking the floxed exon (Fig.S1C), which has several candidate ATG at amino acid 223 and after (Fig.S1C1). We detected a very faint band for smaller molecular weight proteins on western blots, only when the membrane was exposed under 5X longer exposure using Pico PLUS Chemiluminescent Substrate (Thermo Scientific, 34580) and a Licor Odyssey XF Imager (revised Fig. S1B). This smaller molecular weight protein might be produced using any candidate ATGs, but would represent an N-terminal truncated DLK protein lacking the ATP binding site and ~1/4 of the kinase domain, i.e. not a functional kinase.

The revised manuscript has updated citation for DLK(cKO)^fl/fl^. Revised Fig.S1B includes images of a western blot under normal exposure vs longer exposure of western blots using anti-DLK antibodies. New Fig.S1C1 shows effects of floxed exon on DLK.

(2) Why does DLK(OE) not kill CA3 neurons? The phenomenon is clear but there is no link to gene expression changes. In fact, the highlighted transcript in this work, Stmn4, changes in a DLK-dependent manner in CA3.

We agree that this is a very interesting question not answered by our gene expression analysis. While we verified Stmn4 expression levels to correlate to the levels of DLK, we do not think that increased Stmn4 per se in DLK(iOE) is a major factor accounting for CA1 death vs CA3 survival. Several published studies have also reported regulation of Stmn4 mRNAs in other cell types, in the contexts of cell death (Watkins et al., 2013; Le Pichon et al., 2017) and axon regeneration and cytoskeleton disruption (Asghari Adib et al., 2024; DeVault et al., 2024; Hu et al., 2019; Shin et al., 2019). As Stmns have significant expression and function redundancy, conventional knockdown or overexpression of individual Stmn generally does not lead to detectable effects on cellular function. As CA3 neurons are widely known for their dense connections and show resilience to NMDA-mediated neurotoxicity (Sammons et al., 2024; Vornov et al., 1991), we speculate that the differential vulnerability of CA1 and CA3 under DLK(iOE) is a reflection of both the intrinsic property, such as gene expression, and also their circuit connection.

In the revised manuscript, we have included following statement on pg 18:

‘While our data does not pinpoint the molecular changes explaining why CA3 would show less vulnerability to increased DLK, we may speculate that DLK(iOE) induced signal transduction amplification may differ in CA1 vs CA3. CA1 genes appear to be more strongly regulated than CA3 genes, consistent with our observation that increased c-Jun expression in CA1 is greater than that in CA3. Other parallel molecular factors may also contribute to resilience of CA3 neurons to DLK(iOE), such as HSP70 chaperones, different JNK isoforms, and phosphatases, some of which showed differential expression in our RiboTag analysis of DLK(iOE) vs WT (shown in File S2. WT vs DLK(iOE) DEGs). Together with other genes that show dependency on DLK, the DLK and Jun regulatory network contributes to the regional differences in hippocampal neuronal vulnerability under pathological conditions.’

Further we state in ‘Limitation of our study’ on pg 20:

‘Our analysis also does not directly address why CA3 neurons are less vulnerable to increased DLK expression. Future studies using cell-type specific RiboTag profiling and other methods at a refined time window will be required to address how DLK dependent signaling interacts with other networks underlying hippocampal regional neuron vulnerability to pathological insults.’

We hope our data will stimulate continued interests for testable hypothesis in future studies.

(3) Why are whole hippocampi analyzed to IP ribosome-associated mRNAs? The authors nicely show a differential effect of DLK on CA1 vs CA3, but then - at least according to their methods ¬- lyse whole hippocampi to perform IP/sequencing. Their data are therefore a mix of cells where DLK does and does not change cell death. The key issue is whether DLK does/does not have an effect based on the expression changes it drives.

At the time of planning the Ribo-Tag experiment several years ago, we focused on the hippocampal glutamatergic neurons. Due to technical difficulty in micro-dissecting individual hippocampal regions from this early timepoint, we opted to use whole hippocampi to isolate ribosome-associated mRNAs. We agree with the reviewer that it is important to sort out DLK-dependent general gene expression changes vs those specific to a particular cell type where DLK impacts its survival. With emerging CA1, CA3 and other cell-type specific Cre drivers and advanced RNAseq technology, we hope that our work will stimulate broad interest in these questions in future studies.

In the revised manuscript, we have included new analysis comparing our Vglut1-RiboTag profiling (P15) with CamK2-RiboTag (for CA1) and Grik4-RiboTag (for CA3) (P42) published in [83] (GSE209870). We find that >80% of the top ranked genes in their CamK2-RiboTag (for CA1) and Girk4-RiboTag (for CA3) were detected in our VGlut1-RiboTag (revised methods and Supplemental Excel File S3). CA1-enriched genes tended to be expressed higher in DLK(cKO), compared to control, whereas CA3-enriched genes showed less significant correlation to DLK expression levels. Additionally, many genes known to specify CA1 fate do not show significant downregulation in DLK(iOE). This analysis, along with other data in our manuscript, is consistent with an idea that DLK does not regulate neuronal fate.

In the revised manuscript, we presented this additional analysis in Fig. S6K-L, and expanded text description on page 9:

‘Additionally, we compared our Vglut1-RiboTag datasets with CamK2-RiboTag and Grik4-RiboTag datasets from 6-week-old wild type mice reported by (Traunmüller et al., 2023; GSE209870). We defined a list of genes enriched in CamK2-expressing CA1 neurons relative to Grik4-expressing CA3 neurons (CA1 genes), and those enriched in Grik4-expressing CA3 neurons (CA3 genes) (File S3). When compared with the entire list of Vglut1-RiboTag profiling in our control and DLK(cKO), we found CA1 genes tended to be expressed more in DLK(cKO) mice, compared to control (Fig.S6K), while CA3 genes showed a slight enrichment in control though the trend was less significant, and were less clustered towards one genotype (Fig.S6L). Moreover, many CA1 genes related to cell-type specification, such as *FoxP1*, *Satb2*, *Wfs1*, *Gpr161*, *Adcy8*, *Ndst3*, *Chrna5*, *Ldb2*, *Ptpru*, and *Ntm,* did not show significant downregulation when DLK was overexpressed. These observations imply that DLK likely specifically down-regulates CA1 genes both under normal conditions and when overexpressed, with a stronger effect on CA1 genes, compared to CA3 genes. Overall, the informatic analysis suggests that decreased expression of CA1 enriched genes may contribute to CA1 neuron vulnerability to elevated DLK, although it is also possible that the observed down-regulation of these genes is a secondary effect associated with CA1 neuron degeneration’.

(4) Is the subtle decrease in synapse number (Basson/Homer co-loc.) in the DLK (OE) simply a function of neurons (and their synapses, presumably) having died? At the P15 time point that the authors choose because cell death is minimal, there is still a ~25% reduction in CA1 thickness (Figure 2B), which is larger than the ~15% change in synapses (Figure 5H) they describe.

We thank reviewer for the question. To address this, we have analyzed synapses in the CA1 region at P10 in DLK(iOE) mice when there was no detectable loss of neurons. At P10, we did not detect significant changes in Bassoon, Homer1, or colocalized puncta in CA1 (Fig.S11A-F). In P15 DLK(iOE) mice, Homer1 puncta were slightly smaller (Fig.5L) and showed a significant decrease in CA1 SR (Fig.5I).

In the revised manuscript we have also redone our statistical analysis of synapses, using mice rather than ROIs (revised Fig. 5), as recommended by R3. We also analyzed synapses in CA3, and found no significant differences in P10 or P15 (Fig.S12). We would interpret the data to mean that the effects of DLK(OE) on synapses in CA1 may represent an early step in neuronal death. We hope that future studies will shed clarity on this question.

**Reviewer #2 (Public Review):**
This manuscript describes the impact of deleting or enhancing the expression of the neuronal-specific kinase DLK in glutamatergic hippocampal neurons using clever genetic strategies, which demonstrates that DLK deletion had minimal effects while overexpression resulted in neurodegeneration in vivo. To determine the molecular mechanisms underlying this effect, ribotag mice were used to determine changes in active translation which identified Jun and STMN4 as DLK-dependent genes that may contribute to this effect. Finally, experiments in cultured neurons were conducted to better understand the in vivo effects. These experiments demonstrated that DLK overexpression resulted in morphological and synaptic abnormalities.Strengths:This study provides interesting new insights into the role of DLK in the normal function of hippocampal neurons. Specifically, the study identifies:(1) CA1 vs CA3 hippocampal neurons have differing sensitivity to increased DLK signaling.(2) DLK-dependent signaling in these neurons is similar to but distinct from the downstream factors identified in other cell types, highlighted by the identification of STMN4 as a downstream signal.(3) DLK overexpression in hippocampal neurons results in signaling that is similar to that induced by neuronal injury.The study also provides confirmatory evidence that supports previously published work through orthogonal methods, which adds additional confidence to our understanding of DLK signaling in neurons. Taken together, this is a useful addition to our understanding of DLK function.

We thank the reviewer for careful reading and positive comments.

Weaknesses:There are a few weaknesses that limit the impact of this manuscript, most of which are pointed out by the authors in the discussion. Namely:(1) It is difficult to distinguish whether the changes in the translatome identified by the authors are DLK-dependent transcriptional changes, DLK-dependent post-transcriptional changes or secondary gene expression changes that occur as a result of the neurodegeneration that occurs in vivo. Additional expression analysis at earlier time points could be one method to address this concern.

We appreciate the reviewer’s comment, and have performed new analysis on c-Jun and p-c-Jun levels in CA1, CA3, and DG in P10 DLK(OE) mice. Our data suggest that in CA3 elevations in p-c-Jun and c-Jun occur separately from cell death in a DLK-dependent manner, though the high elevation of both p-c-Jun and c-Jun in CA1 correlates with cell death.

The data is presented in revised Fig.S7A,B, and described in revised text on pg 9-10:

‘In control mice, glutamatergic neurons in CA1 had low but detectable c-Jun immunostaining at P10 and P15, but reduced intensity at P60; those in CA3 showed an overall low level of c-Jun immunostaining at P10, P15 and P60; and those in DG showed a low level of c-Jun immunostaining at P10 and P15, and an increased intensity at P60 (Fig.S7A,C,E). In *Vglut1Cre/+;H11-DLKiOE/+* mice at P10 when no discernable neuron degeneration was seen in any regions of hippocampus, only CA3 neurons showed a significant increase of immunostaining intensity of c-Jun, compared to control (Fig.S7A). In P15 mice, we observed further increased immunostaining intensity of c-Jun in CA1, CA3, and DG, with the strongest increase (~4-fold) in CA1, compared to age-matched control mice (Fig.S7C). The overall increased c-Jun staining is consistent with RiboTag analysis.’

Also, on pg.10:

In *Vglut1Cre/+;H11-DLKiOE/+* mice, we observed increased p-c-Jun positive nuclei in CA1 at P10, and strong increase in CA1 (~10-fold), CA3 (~6-fold), and DG (~8-fold) at P15 (Fig.S7B,D).

(2) Related to the above, it is difficult to conclusively determine from the current data whether the changes in synaptic proteins observed in vivo are a secondary result of neuronal degeneration or a primary impact on synapse formation. The in vitro studies suggest this has the potential to be a primary effect, though the difference in experimental paradigm makes it impossible to determine whether the same mechanisms are present in vitro and in vivo.

We appreciate the comment, which is related to R1 point 4. We have performed further analysis and revised the text on pg.12 with the following text:

‘To assess effects of DLK overexpression on synapses, we immunostained hippocampal sections from both P10 and P15, with age-matched littermate controls. Quantification of Bassoon and Homer1 immunostaining revealed no significant differences in CA1 SR and CA3 SR and SL in P10 mice of _<_i>Vglut1^Cre/+^;H11-DLK^iOE/+^ and control (Fig.S11A-F, S12A-J). In P15, Bassoon density and size in CA1 SR were comparable in both mice (Fig 5G, H, K), while Homer1 density and size were reduced in DLK (iOE) (Fig.5G,I, L). Overall synapse number in CA1 SR was similar in DLK (iOE) and control mice (Fig.5J). Similar analysis on CA3 SR and SL detected no significant difference from control (Fig.S12M-V).’

We would interpret the data to mean that the effects of DLK (OE) on synapses in CA1 may represent an early step in neuronal death. We hope that future studies will shed clarity on this question.

Additionally, to address whether the same mechanisms are present in vitro, we have performed further analysis on cultured hippocampal neurons. As described in the Methods, we made hippocampal neuron cultures from P1 pups of the following crosses:

For control: *Vglut1Cre/+* X *Rosa26tdT/+*

For DLKcKO: *Vglut1Cre/+*;DLK(cKO)^fl/fl^ X *Vglut1Cre/+*;DLK(cKO)^fl/fl^;*Rosa26tdT/+*

For DLKiOE: *H11-DLKiOE/iOE* X *Vglut1Cre/+*;*Rosa26tdT/+*

Dissociated cells from a given litter were pooled into the same culture. Because there were different proportions of neurons with our genotype of interest in each culture, it is not simple to know whether DLK was causing significant cell death.

On pg 13, we stated our observation:

‘We did not notice an obvious effect of DLK(iOE) or DLK(cKO) on neuron density in cultures at DIV2. To assess neuronal type distribution in our cultures, we immunostained DIV14 neurons with antibodies for Satb2, as a CA1 marker (Nielsen et al., 2010), and Prox1, as a marker of DG neurons (Iwano et al., 2012). We did not observe significant differences in the proportion of cells labeled with each marker in DLK(cKO) or DLK(iOE) cultures (Fig.S13E). These data are consistent with the idea that DLK signaling does not have a strong role in neuron-type specification both in vivo and in vitro*’*.

(3) The phenotype of DLK cKO mice is very subtle (consistent with previous reports) and while the outcome of increased DLK levels is interesting, the relevance to physiological DLK signaling is less clear. What does seem possible is that increased DLK may phenocopy other neuronal injuries but there are no real comparisons to directly address this in the manuscript. It would be helpful for the authors to provide this analysis as well as a table with all of the translational changes along with fold changes.

Thank you for the suggestion. The fold changes of genes showing significantly altered expression in DLK(cKO) and DLK(iOE) are provided in the excel files (Supplementary excel File S1 WT vs DLK(cKO) DEGs and File S2. WT vs DLK(iOE) DEGs, highlighted columns B and F).

On pg 6, we revised the text as following to include comparison of DLK levels in other physiological conditions and our mice:

‘Several studies have reported that DLK protein levels increase under a variety of conditions, including optic nerve crush (Watkins et al., 2013), NGF withdrawal (~2 fold) (Huntwork-Rodriguez et al., 2013; Larhammar et al., 2017), and sciatic nerve injury (Larhammar et al., 2017). Induced human neurons show increased DLK abundance about ~4 fold in response to ApoE4 treatment (Huang et al., 2019). Increased expression of DLK can lead to its activation through dimerization and autophosphorylation (Nihalani et al., 2000)’.

And,

‘Additional analysis at the mRNA level (supplemental excel, File S2. WT vs DLK(iOE) DEGs) and at the protein level (Fig.S8E) suggest that the increase in DLK abundance was around 3 times the control level. The localization patterns of DLK protein appeared to vary depending on region of hippocampus and age of animals in both control and *Vglut1Cre/+;H11-DLKiOE/+* mice (Fig.S3C).’

In Discussion, we state (pg. 16): ‘The levels of DLK in our DLK(iOE) mice model appear comparable to those reported under traumatic injury and chronic stress.’

(4) For the in vivo experiments, it is unclear whether multiple sections from each animal were quantified for each condition. More information here would be helpful and it is important that any quantification takes multiple sections from each animal into account to account for natural variability.

We apologize this was unclear in the original manuscript.

In the revised methods, under Confocal imaging and quantification (pg 33), we stated: “For brain tissue, three sections per mouse were imaged with a minimum of three mice per genotype for data analysis.”

In revised figure legends, we made it clear that multiple sections from each animal have been used for quantification in all instances, i.e. “Each dot represents averaged thickness from 3 sections per mouse, N≥4 mice/genotype per timepoint.”

In Fig.1F-H: “Each dot represents averaged intensity from 3 sections per mouse”

In Fig.S3B “Data points represent individual mice, averages taken across 3 sections per mouse”

**Reviewer #3 (Public Review):**
Dr Jin and colleagues revisit DLK and its established multifactorial roles in neuronal development, axonal injury, and neurodegeneration. The ambitious aim here is to understand the DLK-dependent gene network in the brain and, to pursue this, they explore the role of DLK in hippocampal glutamatergic neurons using conditional knockout and induced overexpression mice. They produce evidence that dorsal CA1 and dentate gyrus neurons are vulnerable to elevated expression of DLK, while CA3 neurons appear unaffected. Then they identify the DLK-dependent translatome featured by conserved molecular signatures and cell-type specificity. Their evidence suggests that increased DLK signaling is associated with possible STMN4 disruptions to microtubules, among else. They also produce evidence on cultured hippocampal neurons showing that expression levels of DLK are associated with changes in neurite outgrowth, axon specification, and synapse formation. They posit that downstream translational events related to DLK signaling in hippocampal glutamatergic neurons are a generalizable paradigm for understanding neurodegenerative diseases.StrengthsThis is an interesting paper based on a lot of work and a high number of diverse experiments that point to the pervasive roles of DLK in the development of select glutamatergic hippocampal neurons. One should applaud the authors for their work in constructing sophisticated molecular cre-lox tools and their expert Ribotag analysis, as well as technical skill and scholarly treatment of the literature. I am somewhat more skeptical of interpretations and conclusions on spatial anatomical selectivity without stereological approaches and also going directly from (extremely complex) Ribotag profiling patterns to relevance based on immunohistochemistry and no additional interventions to manipulate (e.g. by knocking down or blocking) their top Ribotag profile hits. Also, it seems to this reviewer that major developmental claims in the paper are based on gene translational profiling dependent on DLK expression, not DLK activation, despite some evidence in the paper that there is a correlation between the two. Therefore, observed patterns and correlations may or may not be physiologically or pathologically relevant. Generalizability to neurodegenerative diseases is an overreach not justified by the scope, approach, and findings of the paper.

We thank the reviewer for the encouraging and constructive comments on the manuscript.

Weaknesses and Suggestions:The authors state that the rationale for the translatomic studies is to "to gain molecular understanding of gene expression associated with DLK in glutamatergic neurons" and to characterize the "DLK-dependent molecular and cellular network", However, a problem with the experimental design is the selection of an anatomical region at a time point featured by active neurodegeneration. Therefore, it is not straightforward that the differentially expressed genes or pathways caused by DLK overexpression changes could be due to processes related to neurodegeneration. Indeed, the authors find enrichment of signals related to pathways involved in extracellular matrix organization, apoptosis, unfolded protein responses, the complement cascade, DNA damage responses, and depletion of signals related to mitochondrial electron transport, etc., all of which could be the consequence of neurodegeneration regardless of cause. A more appropriate design to discover DLK-dependent pathways might be to look at a region and/or a time point that is not confounded by neurodegeneration.

We appreciate reviewer’s comment. We included our thoughts in ‘Limitation of the study’ (pg 20):

‘Future studies using cell-type specific RiboTag profiling and other methods at a refined time window will be required to address how DLK dependent signaling interacts with other networks underlying hippocampal regional neuron vulnerability to pathological insults.’

In a related vein, the authors ask "if the differentially expressed genes associated with DLK(iOE) might show correlation to neuronal vulnerability" and, to answer this question, they select the set of differentially expressed genes after DLK overexpression and assess their expression patterns in various regions under normal conditions. It looks to me that this selection is already confounded by neurodegeneration which could be the cause for their downregulation. Therefore, such gene profiles may not be directly linked to neuronal vulnerability. A similar issue also relates to the conclusion that "...the enrichment of DLK-dependent translation of genes in CA1 suggests that the decreased expression of these genes may contribute to CA1 neuron vulnerability to elevated DLK".

We agree with the reviewer’s concern that it is difficult to separate neurodegenerative consequences from changes caused by DLK solely based on our translatomics studies on P15 DLK(iOE) mice. As responded to reviewer 1 (point 4) and reviewer 2 (point 1), we have included new analysis of P10 mice (Fig.S7A,B) when neurons did not show detectable sign of degeneration.

We consider several lines of evidence supporting that some differentially expressed genes in DLK(iOE) vs control may likely be specific for increased DLK signaling.

First, the genes identified in DLK(iOE) vs control represent a small set of genes (260), which is comparable to other DLK dependent datasets (Asghari Adib et al., 2024) but shows cell-type specificity.

Second, our analysis using rank-rank hypergeometric overlap (RRHO) detects a significant correlation between upregulated genes from DLK(iOE) vs downregulated genes in DLK(cKO), and vice versa, suggesting that expression of a similar set of genes is depended on DLK (Fig.3C, S6C-E). Consistently, GO term analysis using the list of genes coordinately regulated by DLK, derived from our RRHO analysis, leads to identification of similar GO terms related to up- and downregulated genes as using DLK(iOE)-RiboTag data alone. SynGO analysis of DLK(iOE) regulated genes and DLK(cKO) regulated genes also identified similar synaptic processes regulated by significantly regulated genes (Fig.3F and S6J).

Third, we performed additional analysis comparing our Vglut1-RiboTag dataset with CamK2-RiboTag and Grik4-RiboTag datasets from 6-week-old wild type mice reported by (Traunmüller et al., 2023; GSE209870). We observed >80% overlap among the top ranked genes (revised Methods). We described this analysis on pg 9 and Fig. S6K-L (and Supplemental Excel File S3):

‘Additionally, we compared our Vglut1-RiboTag datasets with CamK2-RiboTag and Grik4-RiboTag datasets from 6-week-old wild type mice reported by (Traunmüller et al., 2023; GSE209870). We defined a list of genes enriched in CamK2-expressing CA1 neurons relative to Grik4-expressing CA3 neurons (CA1 genes), and those enriched in Grik4-expressing CA3 neurons (CA3 genes) (File S3). When compared with the entire list of Vglut1-RiboTag profiling in our control and DLK(cKO), we found CA1 genes tended to be expressed more in DLK(cKO) mice, compared to control (Fig.S6K), while CA3 genes showed a slight enrichment in control though the trend was less significant, and were less clustered towards one genotype (Fig.S6L). Moreover, many CA1 genes related to cell-type specification, such as *FoxP1*, *Satb2*, *Wfs1*, *Gpr161*, *Adcy8*, *Ndst3*, *Chrna5*, *Ldb2*, *Ptpru*, and *Ntm,* did not show significant downregulation when DLK was overexpressed. These observations imply that DLK likely specifically down-regulates CA1 genes both under normal conditions and when overexpressed, with a stronger effect on CA1 genes, compared to CA3 genes. Overall, the informatic analysis suggests that decreased expression of CA1 enriched genes may contribute to CA1 neuron vulnerability to elevated DLK, although it is also possible that the observed down-regulation of these genes is a secondary effect associated with CA1 neuron degeneration.’

To understand the role and relevance of the DLK overexpression model, there should be a discussion of to what extent it corresponds to endogenous levels of DLK expression or DLK-MAPK pathway activation under baseline or pathological conditions.

We appreciate the suggestion, which is similar to R2 point 3. We have revised the text and discussion to include how DLK levels may be altered in other physiological conditions vs our mice.

Pg. 6: ‘Several studies have reported that DLK protein levels increase under a variety of conditions, including optic nerve crush (Watkins et al., 2013), NGF withdrawal (~2 fold) (Huntwork-Rodriguez et al., 2013; Larhammar et al., 2017), and sciatic nerve injury (Larhammar et al., 2017). Induced human neurons show increased DLK abundance about ~4 fold in response to ApoE4 treatment (Huang et al., 2019). Increased expression of DLK can lead to its activation through dimerization and autophosphorylation (Nihalani et al., 2000)’.

And,

‘Additional analysis at the mRNA level (supplemental excel, File S2. WT vs DLK(iOE) DEGs) and at the protein level (Fig.S8E) suggest that the increase in DLK abundance was around 3 times the control level. The localization patterns of DLK protein appeared to vary depending on region of hippocampus and age of animals in both control and *Vglut1Cre/+;H11-DLKiOE/+* mice (Fig.S3C).’

In Discussion (pg. 16): ‘The levels of DLK in our DLK(iOE) mice model appear comparable to those reported under traumatic injury and chronic stress.’

The authors posit that "dorsal CA1 neurons are vulnerable to elevated DLK expression, while neurons in CA3 appear largely resistant to DLK overexpression". This statement assumes that DLK expression levels start at a similar baseline among regions. Do the authors have any such data? Ideally, they should show whether DLK expression and p-c-Jun (as a marker of downstream DLK signaling) are the same or different across regions in both WT and overexpression mice. For example, what are the DLK/p-c-Jun expression levels in regions other than CA1 in Supplementary Figures 2-3 and how do they compare with each other? Normalization to baseline for each region does not allow such a comparison. Also, in Supplementary Figure 6, analyses and comparisons between regions are done at a time point when degeneration has already started. Ideally, these should be done at P10.

We thank the reviewer for raising these points. In the revised manuscript we have included protein expression analysis of DLK (Fig S3), c-Jun, and p-c-Jun at P10 (Fig. S7).

We provided a quantification of DLK immunostaining intensity in CA1 and CA3 in Fig.S3D,E and find roughly comparable levels between regions.

Pg. 6: ‘Additional analysis at the mRNA level (supplemental excel, File S2. WT vs DLK(iOE) DEGs) and at the protein level (Fig.S8E) suggest that the increase in DLK abundance was around 3 times the control level. The localization patterns of DLK protein appeared to vary depending on region of hippocampus and age of animals in both control and *Vglut1Cre/+;H11-DLKiOE/+* mice (Fig.S3C).’

We provided our quantifications without normalization to baseline in each region for c-Jun and p-c-Jun, and revised the text accordingly:

Pg. 9-10: ‘In control mice, glutamatergic neurons in CA1 had low but detectable c-Jun immunostaining at P10 and P15, but reduced intensity at P60; those in CA3 showed an overall low level of c-Jun immunostaining at P10, P15 and P60; and those in DG showed a low level of c-Jun immunostaining at P10 and P15, and an increased intensity at P60 (Fig.S7A,C,E). In *Vglut1Cre/+;H11-DLKiOE/+* mice at P10 when no discernable neuron degeneration was seen in any regions of hippocampus, only CA3 neurons showed a significant increase of immunostaining intensity of c-Jun, compared to control (Fig.S7A). In P15 mice, we observed further increased immunostaining intensity of c-Jun in CA1, CA3, and DG, with the strongest increase (~4-fold) in CA1, compared to age-matched control mice (Fig.S7C). The overall increased c-Jun staining is consistent with RiboTag analysis’.

Pg. 10: ‘In *Vglut1Cre/+;H11-DLKiOE/+* mice, we observed increased p-c-Jun positive nuclei in CA1 at P10, and strong increase in CA1 (~10-fold), CA3 (~6-fold), and DG (~8-fold) at P15 (Fig.S7B,D).

Illustration of proposed selective changes in hippocampal sector volume needs to be very carefully prepared in view of the substantial claims on selective vulnerability. In 2A under P15 and especially P60, it is difficult to see the difference - this needs lower magnification and a lot of care that anteroposterior levels are identical because hippocampal sector anatomy and volumes of sectors vary from level to level. One wonders if the cortex shrinks, too. This is important.

Thank you for raising the point. We have provided images to view the anteroposterior level in Fig.S2A-C. We have noticed cortex in DLK(OE) mice to become thinner, along with expansion of ventricles in some animals at later timepoints (Fig.S2C).

One cannot be sure that there is selective death of hippocampal sectors with DLK overexpression versus, say, rearrangement of hippocampal architecture. One may need stereological analysis, otherwise this substantial claim appears overinterpreted.

We appreciate the comment.

In the revised manuscript, we included a new supplemental figure (Fig. S2) showing lower magnification images of coronal sections, and used cautionary wording, such as ‘CA3 is less vulnerable, compared to CA1’, to minimize the impression of over-interpretation. By NeuN staining, at P10, P15, P60, we did not observe detectable difference in overall hippocampus architecture, apart from noted cell death of CA1 and DG and associated thinning of each of the layers. At 46 weeks, some animals showed differences in the overall shape of dorsal hippocampus, though this appeared to reflect a disproportionately large CA3 region compared to other regions (Fig S2). Increased GFAP staining (Fig.S5A-C) was detected in CA1 but not in CA3, and microglia by IBA1 staining (Fig.S5E) also displayed less reactivity in CA3, compared to CA1. Thus, based on NeuN staining, GFAP staining, IBA1 staining and analysis of the differentially regulated genes, we infer that the effect of DLK(iOE) in CA1 is different than the effect on CA3.

Is the GFAP excess reflective of neuroinflammation? What do microglial markers show? The presence of neuroinflammation does not bode well with apoptosis. Speaking of which, TUNEL in one cell in Supplementary Figure 4E is not strong evidence of a more widespread apoptotic event in CA1.

We have included staining data for the microglia marker IBA1. Both GFAP and IBA1 showed evidence of reactivity particularly in the CA1 region (S5A-E), supporting the differential vulnerability in different regions, though whether cell death is primarily due to apoptosis is unclear.

We agree that our data of sparse TUNEL staining at P15 (Fig S5F,G) do not rule out whether other mechanisms of cell death may also occur. We have included this in our limitations (pg.20) “While we find evidence for apoptosis, other forms of cell death may also occur.”

In several places in the paper (as illustrated in Figure 4B, Supplementary Figure 2B, etc.): the unit of biological observation in animal models is typically not a cell, but an organism, in which averaged measures are generated. This is a significant methodological problem because it is not easy to sample neurons without involving stereological methods. With the approach taken here, there is a risk that significance may be overblown.

We appreciate the reviewer’s point. We used same region for quantification of RNAscope, genotype-blind when possible. We revised the graphs to show mean values for individual mice in Fig.4B, 4C, and Fig.S3B (previously Fig.S2B).

Other Comments and Questions:Supplementary Figure 9: The authors state that data points are shown for individual ROIs - ideally, they should also show averages for biological replicates. Can the authors confirm that statistical analyses are based on biological replicates (mice) and not ROIs?

We have revised the graphs to show averages from individual mice in Fig.5B-D, F5E-F (previously Fig.S9G-I), Fig.5H-J, and Fig.5K-L (previously Fig.S9J-L) and Fig.S10B,C,E,F (previously Fig.S9B,C, E,F). The statistical analyses are based on biological replicates of mice.

For in vitro experiments, what is the effect of DLK overexpression on neuronal viability and density? Could these variables confound effects on synaptogenesis/synapse maturation?

As described in the Methods, we made hippocampal neuron cultures from P1 pups of the following crosses:

For control: *Vglut1Cre/+* X *Rosa26tdT/+*

For DLKcKO: *Vglut1Cre/+*;DLK(cKO)^fl/fl^ X *Vglut1Cre/+*;DLK(cKO)^fl/fl^;*Rosa26tdT/+*

For DLKiOE: *H11-DLKiOE/iOE* X *Vglut1Cre/+*;*Rosa26tdT/+*

Dissociated cells from a given litter were pooled into the same culture. Because there were different proportions of neurons with our genotype of interest in each culture, it is not simple to know whether DLK was causing significant cell death.

On pg 13, we stated our observation:

‘We did not notice an obvious effect of DLK(iOE) or DLK(cKO) on neuron density in cultures at DIV2. To assess neuronal type distribution in our cultures, we immunostained DIV14 neurons with antibodies for Satb2, as a CA1 marker (Nielsen et al., 2010), and Prox1, as a marker of DG neurons (Iwano et al., 2012). We did not observe significant differences in the proportion of cells labeled with each marker in DLK(cKO) or DLK(iOE) cultures (Fig.S13E). These data are consistent with the idea that DLK signaling does not have a strong role in neuron-type specification both in vivo and in vitro*’*.

We cannot rule out whether variable factors in our cultures may confound effects on synaptogenesis/synapse maturation, and would hope future studies will shed clarity.

Correlations between c-jun expression and phosphorylation are extremely important and need to be carefully and convincingly documented. I am a bit concerned about Supplementary Figure 6 images, especially 6B-CA1 (no difference between control and KO, too small images) and 6D (no p-c-Jun expression at all anywhere in the hippocampus at P15?).

At P10, P15, and P60 we stained for p-c-Jun using the Rabbit monoclonal p-c-Jun (Ser73) (D47G9) antibody from Cell Signaling (cat# 3270) at a 1:200 dilution and imaged using an LSM800 confocal microscope with a 20x objective. We observed p-c-Jun to be quite low generally in control animals. We have replaced the images in Fig.S7F (previously S6D), and adjusted the brightness/contrast to enable better visualization of the low signal in Fig.S7B,D,F (previously Fig.S6B,D).

We revised our text to present the data carefully as stated above:

Pg. 9-10: ‘In control mice, glutamatergic neurons in CA1 had low but detectable c-Jun immunostaining at P10 and P15, but reduced intensity at P60; those in CA3 showed an overall low level of c-Jun immunostaining at P10, P15 and P60; and those in DG showed a low level of c-Jun immunostaining at P10 and P15, and an increased intensity at P60 (Fig.S7A,C,E). In *Vglut1Cre/+;H11-DLKiOE/+* mice at P10 when no discernable neuron degeneration was seen in any regions of hippocampus, only CA3 neurons showed a significant increase of immunostaining intensity of c-Jun, compared to control (Fig.S7A). In P15 mice, we observed further increased immunostaining intensity of c-Jun in CA1, CA3, and DG, with the strongest increase (~4-fold) in CA1, compared to age-matched control mice (Fig.S7C). The overall increased c-Jun staining is consistent with RiboTag analysis’.

Pg. 10: ‘In *Vglut1Cre/+;H11-DLKiOE/+* mice, we observed increased p-c-Jun positive nuclei in CA1 at P10, and strong increase in CA1 (~10-fold), CA3 (~6-fold), and DG (~8-fold) at P15 (Fig.S7B,D).

**Recommendations for the authors:**
Several major and minor reservations were raised. The major issues are the need for more information about the over-expression of DLK and a need to extrapolate to an in vivo condition with DLK. A considerable amount of useful information is presented with some very nicely done experiments but it is not yet a coherent or integrated story. The lack of impact of DLK overexpression in some neurons is perhaps the most impactful observation of the study and would be great to have more information around the differential transcriptional/signaling response in these cell types. There is also a need for more experimental details and to address several questions about the mouse genetic and translatome analysis. They are valid concerns that require attention by the authors.

We thank the editors and reviewers for their thoughtful evaluation and suggestions. We hope that the editors and reviewers find that the new data and text changes in our revised manuscript, along with above point-to-point response, have addressed the concerns and strengthened our findings.

Minor points:(1)The authors state that deletion of DLK has no effect on CA1 at 1yr, however, the image of CA1 in Figure S1D shows substantially fewer NeuN+ neurons. Is this a representative field of view?

We have re-examined images, and observed no effect on hippocampal morphology at 1 yr. We now included representative images in the revised Fig S1D.

(2) Is the DLK protein section staining in Figure 2C a real signal? The staining looks like speckles and is purely somatic. Axonal staining is widely expected based on the literature and the authors' own work. There should be a specificity control.

To our knowledge, axonal staining of DLK reported in the literature is mostly based on cultured DRG neurons. In addition to the reported axonal localization, DLK is present in the cell soma, near the golgi (Hirai et al., 2002), and in the post-synaptic density (Pozniak et al., 2013).

In the revised manuscript, we addressed this point by including controls with no primary antibody, and using an antibody against the closely related kinase, LZK. These additional data are shown in (Fig.S3C,D) (previously Fig.S2C), supporting that DLK protein staining represents real signal. At P10 and P15, DLK immunostaining around CA3 showed axonal staining of the mossy fibers, as well as in the soma and dendritic layers (Fig.S3C,D). A similar pattern was also seen in primary cultured neurons (Fig 6A).

(3) The protein expression of DLK in the transgenic overexpressor (Figure S7C) looks, to the resolution of this blot, to be at least 50kD heavier than 'WT' DLK. Can the authors explain this discrepancy?

The Cre-induced DLK(iOE) transgene has T2A and tdTomato in-frame to C-terminus of DLK. It is known that T2A ‘self-cleavage’ is often incomplete. DLK-T2A-tdTomato would be about 50 kD bigger than WT DLK. We now include the transgene design in revised Fig S1D, and also stated in figure legend of Fig.S8C (previously S7C) that ‘Larger molecular weight band of DLK in *Vglut1Cre/+*;H11-DLKiOE/+ would match the predicted molecular weight of DLK-T2A-tdTomato if T2A-peptide induced ‘self-cleavage’ due to ribosomal skipping is ineffective (Fig.S1D).’

(4) Expression changes in DLK affect various aspects of neurites in CA1 cultures (Figure 6), and changes in DLK also modestly affect STMN4 (and 2, perhaps indirectly) levels (Figure S7C), but there is no indication that DLK acts via STMN4 to cause these changes. It is not clear what to make of these data. Of note, Stmn4 levels change in response to DLK in CA3, without DLK affecting cell death in this region.

We appreciate and agree with the comment. Other studies (Asghari Adib et al., 2024; DeVault et al., 2024; Hu et al., 2019; Larhammar et al., 2017; Le Pichon et al., 2017; Shin et al., 2019; Watkins et al., 2013) reported expression changes in Stmn4 mRNAs in other cell types and cellular contexts, which appeared to depend on DLK. Hippocampal neurons express multiple Stmns (Fig.S8A). While we present our analysis on the effects of DLK dosage on Stmn4, and also Stmn2, we do not think that DLK-induced changes of Stmn4 expression per se is a major factor underlying CA1 cell death vs CA3 survival.

In the revised manuscript, we addressed this point in ‘Limitation of our study’ (pg 20):

‘Additional experiments will be needed to elucidate in vivo roles of STMN4 and its interaction with other STMNs’.